# Simultaneous two-photon imaging and two-photon optogenetics of cortical circuits in three dimensions

**Weijian Yang\*, Luis Carrillo-Reid, Yuki Bando, Darcy S Peterka, Rafael Yuste\***

NeuroTechnology Center, Department of Biological Sciences, Columbia University, New York, United States

**Abstract** The simultaneous imaging and manipulating of neural activity could enable the functional dissection of neural circuits. Here we have combined two-photon optogenetics with simultaneous volumetric two-photon calcium imaging to measure and manipulate neural activity in mouse neocortex in vivo in three-dimensions (3D) with cellular resolution. Using a hybrid holographic approach, we simultaneously photostimulate more than 80 neurons over 150 μm in depth in layer 2/3 of the mouse visual cortex, while simultaneously imaging the activity of the surrounding neurons. We validate the usefulness of the method by photoactivating in 3D selected groups of interneurons, suppressing the response of nearby pyramidal neurons to visual stimuli in awake animals. Our all-optical approach could be used as a general platform to read and write neuronal activity.
DOI: https://doi.org/10.7554/eLife.32671.001

## Introduction

The precise monitoring and control of neuronal activity may be an invaluable tool to decipher the function of neuronal circuits. For reading out neuronal activity in vivo, the combination of calcium imaging of neuronal populations (*Yuste and Katz, 1991*) with two-photon microscopy (*Denk et al., 1990*), has proved its utility because of its high selectivity, good signal-to-noise ratio, and depth penetration in scattering tissues (*Yuste and Denk, 1995*; *Cossart et al., 2003*; *Zipfel et al., 2003*; *Helmchen and Denk, 2005*; *Svoboda and Yasuda, 2006*; *Ji et al., 2016*; *Yang and Yuste, 2017*). Moreover, two-photon imaging can be combined with two-photon optochemistry (*Nikolenko et al., 2007*; *Dal Maschio et al., 2010*) or two-photon optogenetics (*Packer et al., 2012*; *Rickgauer et al., 2014*; *Packer et al., 2015*; *Emiliani et al., 2015*; *Carrillo-Reid et al., 2016*) to allow simultaneous readout and manipulation of neural activity with cellular resolution. But so far, the combinations of these optical methods into an all-optical approach have been largely restricted to two-dimensional (2D) planes (*Nikolenko et al., 2007*; *Dal Maschio et al., 2010*; *Rickgauer et al., 2014*; *Packer et al., 2015*; *Carrillo-Reid et al., 2016*). At the same time, neural circuits are three dimensional, and neuronal sub-populations are distributed throughout their volume. Therefore, extending these methods to three dimensions (3D) appears essential to enable systematic studies of microcircuit computation and processing.

Here we employed wavefront shaping strategies with a customized dual-beam two-photon microscope to simultaneously perform volumetric calcium imaging and 3D patterned photostimulations in mouse cortex in vivo. For phostostimulation, we adopted a hybrid strategy that combines 3D holograms and galvanometer driven spiral scans. Furthermore, we used a pulse-amplified low-repetition-rate (200 kHz ~ 1 MHz) laser, which significantly reduces the average laser power required for photoactivation, and minimizes thermal effects and imaging artifacts. With this system, we photostimulated large groups of cells simultaneously in layer 2/3 of primary visual cortex (V1) in awake mice

**\*For correspondence:**
wejyang@ucdavis.edu (WY);
rmy5@columbia.edu (RY)

**eLife digest** Modern microscopy provides a window into the brain. The first light microscopes were able to magnify cells only in thin slices of tissue. By contrast, today's light microscopes can image cells below the surface of the brain of a living animal. Even so, this remains challenging for several reasons. One is that the brain is three-dimensional. Another is that brain tissue scatters light. Trying to view neurons deep within the brain is a little like trying to view them through a glass of milk. Most of the light scatters on its way through the tissue with the result that little of the light reaches the target neurons.

Yang et al. have now tackled these challenges using a technique called holography. Holography produces 3D images of objects by splitting a beam of light and then recombining the beams in a specific way. Yang et al. applied this technique to an infrared laser beam, opting for infrared because it scatters much less in brain tissue than visible light. Directing each of the infrared beams to a different neuron can produce 3D images of multiple cells within the brain's outer layer, the cortex, all at the same time.

The holographic infrared microscope can be used alongside two techniques called optogenetics and calcium imaging, in which light-sensitive proteins are inserted into neurons. Depending on the proteins introduced, shining light onto the neurons will either change their activity, or cause them to fluoresce whenever they are active. Just as a computer can both "read" and "write" data, the holographic microscope can thus read out existing neuronal activity or write new patterns of activity. By combining these techniques, Yang et al. were able to stimulate more than 80 neurons at the same time – and meanwhile visualize the activity of the surrounding neurons – at multiple depths within the mouse cortex.

This new microscopy technique, while a clear advance over existing methods, still cannot image and control neurons throughout the entire cortex. The next goal is to further extend this method across multiple brain areas and manipulate the activity of any subset of neurons at will. Neuroscientists will greatly benefit from the ability to image and alter the activity of living neural circuits in 3D. In the future, clinicians may be able to use this technique to treat brain disorders by adjusting the activity of abnormal neural circuits.

DOI: https://doi.org/10.7554/eLife.32671.002

(>80 cells distributed within a $480 \times 480 \times 150 \ \mu m^3$ imaged volume), while simultaneously imaging the activity of the surrounding neurons. Compared with other 3D all-optical approaches (*Dal Maschio et al., 2017*; *Mardinly et al., 2017*), which used scanless holographic photostimulation, our hybrid approach requires less laser power to stimulate per cell, and can thus simultaneously photostimulate more cells for a given fixed power budget.

This all-optical method is useful to analyze the function of neural circuits in 3D, such as studying cell connectivity, ensemble organization, information processing, or excitatory and inhibitory balance. As a demonstration, we photostimulated groups of pyramidal cells in 3D with high specificity, and also targeted a selective population of interneurons in V1 in awake mice, finding that stimulating the interneurons reduced the response of pyramidal cells to visual simuli.

## Results

We built a holographic microscope with two independent two-photon lasers, one for imaging and the other for photostimulation (*Figure 1A*). Each laser beam's axial focal depth could be controlled without mechanical motion of the objective, yielding maximum flexibility while reducing perturbations to the animal. On the imaging path, we coupled a wavelength-tunable Ti:Sapphire laser through an electrically tunable lens (ETL, EL-10-30-C-NIR-LD-MV, Optotune AG) (*Grewe et al., 2011*) followed by a resonant scanner for high speed volumetric imaging. The ETL, as configured, provided an adjustable axial focus shift up to 90 μm below and 200 μm above the objective's nominal focal plane. On the photostimulation path, we used a low-repetition-rate ultrafast laser coupled to a spatial light modulator (SLM, HSP512-1064, Meadowlark Optics) to shape the wavefront, allowing flexible 3D beam splitting that simultaneously targets the user defined positions in the sample (*Figure 1B–1E*). The axial and lateral targeting error was 0.59 ± 0.54 μm and 0.82 ± 0.65 μm,

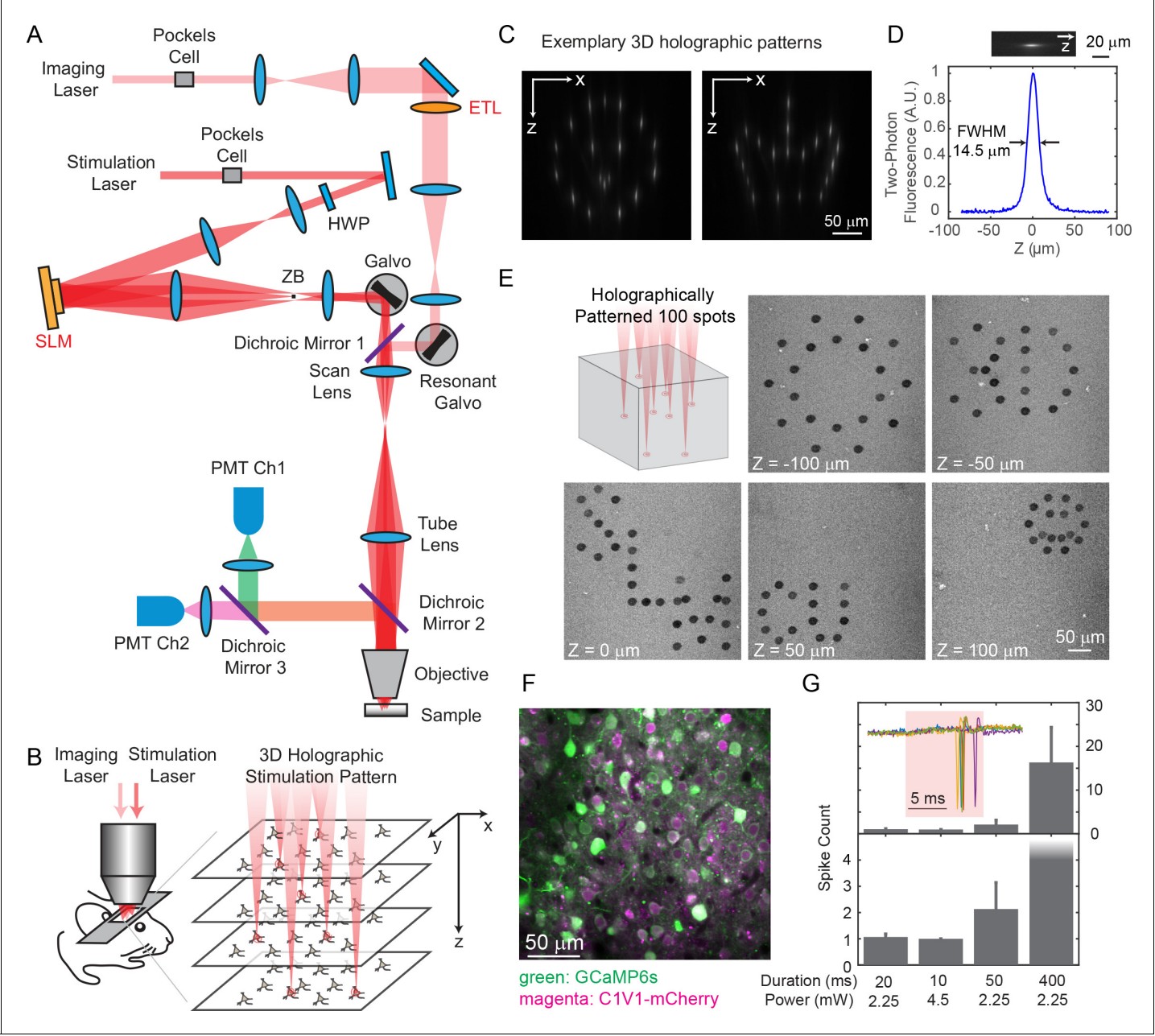

**Figure 1.** Two-photon imaging and photostimulation microscope. (**A**) Dual two-photon excitation microscope setup. HWP, half-wave plate; ZB, zeroth-order beam block; SLM, spatial light modulator; ETL, electrically tunable lens; PMT, photomultiplier tube. (**B**) Schematics for simultaneous volumetric calcium imaging and 3D holographic patterned photostimulation in mouse cortex. (**C**) Exemplary 3D holographic patterns projected into Alexa 568 fluorescence liquid with its xz cross section captured by a camera. (**D**) Measured point spread function (PSF) in the axial (z) direction for two-photon excitation (photostimulation path). The full-width-at-half-maximum (FWHM) is 14.5 μm, corresponding to an NA ~ 0.35. (**E**) 100 spots holographic pattern spirally scanned by a post-SLM galvanometric mirror bleaching an autofluorescence plastic slide across five different planes. (**F**) A typical field of view showing neurons co-expressing GCaMP6s (green) and C1V1-mCherry (magenta). (**G**) Spike counts of target pyramidal cells in layer 2/3 of mouse V1 evoked by photostimulation with different spiral duration and average laser power (3 cells in each condition; mice anesthetized; 1 MHz repetition rate for photostimulation laser). The inset shows the cell-attached recording of a 10 ms spiral stimulation over five trials in a neuron. The red shaded area indicates photostimulation period. Error bars are standard error of the mean over cells.

DOI: https://doi.org/10.7554/eLife.32671.003

The following figure supplements are available for figure 1:

**Figure supplement 1.** System characterization of the spatial light modulator (SLM) in the 3D microscope.

DOI: https://doi.org/10.7554/eLife.32671.004

*Figure 1 continued on next page*

*Figure 1 continued*

**Figure supplement 2.** Characterization and spatial resolution of photostimulation.
DOI: https://doi.org/10.7554/eLife.32671.005
**Figure supplement 3.** Cross talk from imaging into photostimulation.
DOI: https://doi.org/10.7554/eLife.32671.006
**Figure supplement 4.** Cross talk from photostimulation into imaging.
DOI: https://doi.org/10.7554/eLife.32671.007

respectively, across a 3D field of view (FOV) of 240 × 240 × 300 µm³ (*Figure 1—figure supplement 1*; Materials and Methods). The SLM path was coupled through a pair of standard galvanometers that can allow for fast extension of the targeting FOV beyond that nominal addressable SLM-only range (*Yang et al., 2015*). For optogenetics experiments, we actuated this pair of galvanometric mirrors to scan the beamlets in a spiral over the cell bodies of the targeted neuron (see *Figure 1E* for an exemplary 3D pattern with 100 targets on an autofluorescent plastic slide). We term this a 'hybrid' approach, as it combined holography with mechanical scanning, as opposed to purely holographic approach. For in vivo experiments, we imaged green fluorescence from the genetically encoded calcium indicator GCaMP6s or GCaMP6f (*Chen et al., 2013*) and photostimulated a red-shifted opsin, C1V1-mCherry (*Yizhar et al., 2011*). With switchable kinematic mirrors and dichroic mirrors, the lasers could be easily redirected to whichever path, and thus the system could also be utilized for red fluorophores and blue opsins.

We co-expressed GCaMP6s or GCaMP6f (*Chen et al., 2013*) and C1V1-p2A (*Yizhar et al., 2011*) in mouse V1 (*Figure 1F*), and excited them with 940 nm and 1040 nm light, respectively. The separation of their excitation spectrum allowed for minimal cross-talk between the imaging and photostimulation paths (Discussion). C1V1-expressed cells were identified through a co-expressed mCherry fluorophore. Single spikes can be evoked with very low average laser power (~2.25 mW with 20 ms spiral, or ~ 4.5 mW with 10 ms spiral, 1 MHz pulse train, layer 2/3 in vivo, *Figure 1G*), latency and jitter (17.0 ± 4.2/8.5 ± 1.6 ms latency, and 2.0±1.5/0.5±0.3 ms jitter for the two conditions, *Figure 1—figure supplement 2*; jitter defined as the standard deviation of the latency). With a higher power (10 ~ 20 mW), neural activity could also be evoked with photostimulation duration as short as 1 ms (*Figure 2*).

Compared with alternative scanless strategy like temporal focusing (*Rickgauer et al., 2014*; *Mardinly et al., 2017*; *Hernandez et al., 2016*; *Pégard et al., 2017*) or pure holographic approaches (*Dal Maschio et al., 2017*), where the laser power is distributed across the whole cell body of each targeted neuron, our hybrid approach is simple, accommodates large numbers of simultaneous targets, and appears to have a better power budget for large population photostimulation in general. To test this, we compared the required power budget for hybrid approach and the scanless (pure holographic) approach at different photostimulation durations (20 ms, 10 ms, 5 ms and 1 ms). On our system, when photostimulation duration was above 5 ms, the hybrid approach required about half of the laser power than the scanless approach to evoke similar response in the neuron; at 1 ms photostimulation duration, the hybrid approach shows a trend with smaller power budget (but not significant, p=0.17 using one-way ANOVA test) than the scanless approach (*Figure 2*, *Figure 2—figure supplement 1*). One reason for this difference is that the scanless approach employs a spatial multiplexed strategy, where the two-photon light is spatially distributed across the entire cell body; to maintain the two-photon excitation efficiency (squared-intensity) within its coverage area, a larger total power is typically required. The hybrid approach, on the other hand, is a combination of spatial (across different cells) and temporal (within individual cell) multiplexed strategy. While optimal strategy will depend on opsin photophysics, the opsin typically has a long opsin decay constant (*Mattis et al., 2011*) (10 s of millisecond) and this favors the hybrid approach because the opsin channels can stay open during the entire (multiple) spiral scans. But at very short duration, the limited number of laser pulses per unit area may contribute to an efficiency drop of the hybrid approach versus scanless approach.

We tested our 3D all-optical system by targeting and photoactivating selected groups of pyramidal cells throughout three axial depths of layer 2/3 of V1 in anesthetized mice, while simultaneously monitoring neuronal activity in those three planes (240 × 240 µm² FOV for each plane) at 6.67 vol/s. Neurons were photoactivated one at a time (*Figure 3—figure supplement 1*), or as groups/

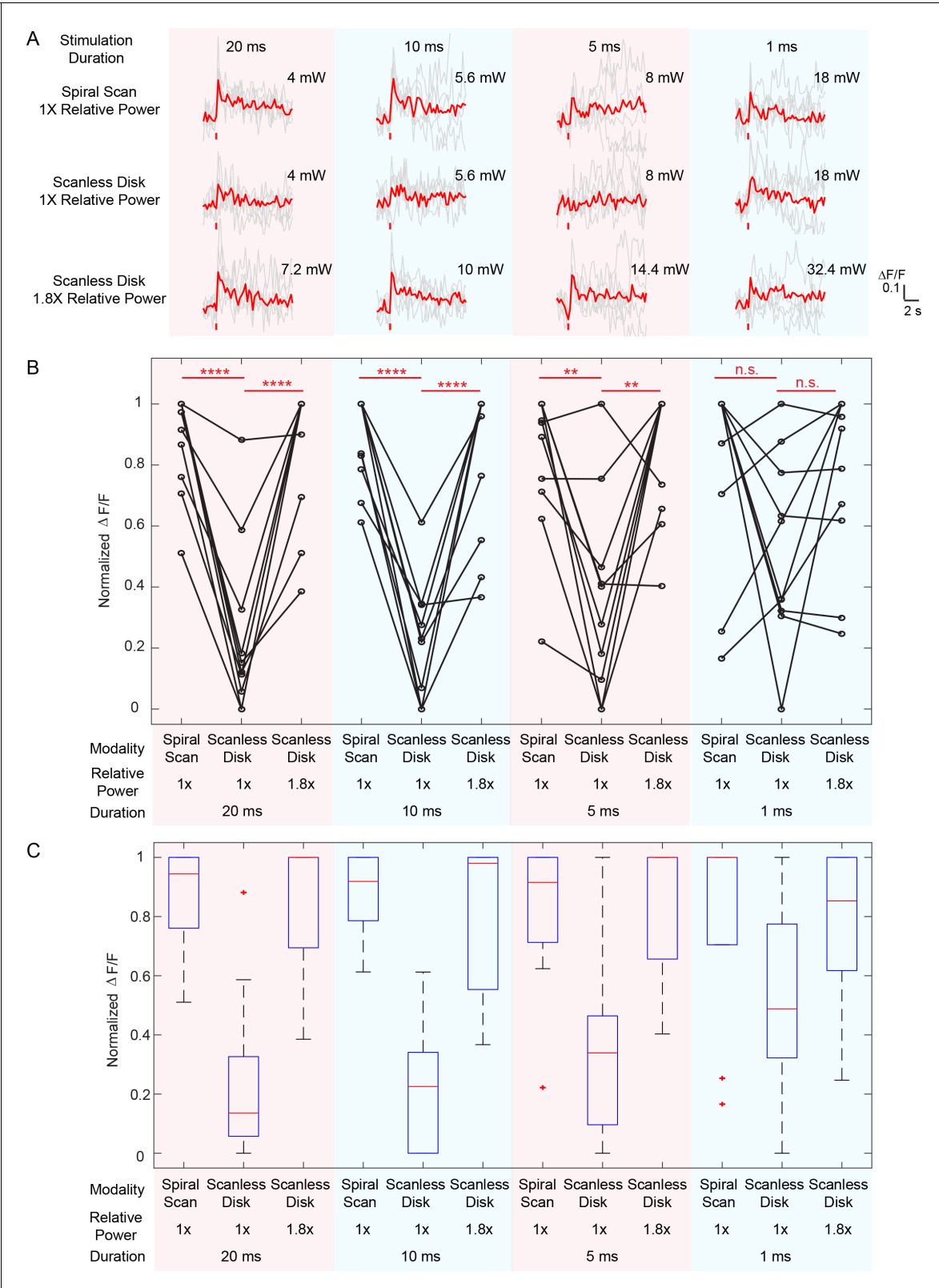

**Figure 2.** Comparison between spiral scan and scanless holographic approaches for photostimulation. In the scanning approach, the laser spot is spirally scanned over the cell body; in the scanless approach, a disk pattern (~12 μm in diameter) is generated by the SLM, covering the entire cell body at once. (**A**) Photostimulation triggered calcium response of a targeted neuron in vivo at mouse layer 2/3 of V1, for different stimulation modalities. For each modality, the multiplication of stimulation duration and the power squared was kept constant over four different stimulation durations. The

*Figure 2 continued on next page*

*Figure 2 continued*

average response traces are plotted over those from the individual trials. (B) ΔF/F response of neurons on different photostimulation conditions [10 cells over two mice in vivo (photostimulated one at a time), layer 2/3 of V1, over a depth of 100 ~ 270 µm from pial surface; one-way ANOVA test show significant different response between spiral scan and scanless approach at the same power for stimulation duration of 20 ms, 10 ms and 5 ms. At 1 ms, the p value is 0.17]. For each neuron and each stimulation duration, the power used in the scanless disk modality is 1 and 1.8 times relative to that in the spiral scan. For each neuron and each modality, the multiplication of the stimulation duration and the power squared was kept constant over four different stimulation durations. The power used in the spiral scan with 20 ms duration varies from 2.2 mW to 5 mW for different cells. (C) Boxplot summarizing the statistics in (B). The central mark indicates the median, and the bottom and top edges of the box indicate the 25th and 75th percentiles, respectively. The whiskers extend to the most extreme data points (99.3% coverage if the data are normal distributed) not considered outliers, and the outliers are plotted individually using the '+' symbol. In this experiment, the mice are transfected with GCaMP6f and C1V1-mCherry. Repetition rate of the photostimulation laser is 1 MHz. The spiral scan consists of 50 rotations with progressively shrinking radius, and the scanning speed is adjusted to make different stimulation durations.

DOI: https://doi.org/10.7554/eLife.32671.008

The following figure supplement is available for figure 2:

**Figure supplement 1.** Comparison between spiral scan and scanless holographic approaches for photostimulation.

DOI: https://doi.org/10.7554/eLife.32671.009

ensembles (*M* neurons simultaneously, *M* = 3 ~ 27, *Figure 3*) and the majority of the targeted cells (86 ± 6%, Materials and Methods) showed clear calcium transients in response to the photostimulation (*Figure 3C–E*).

We further investigated the reliability of the photoactivation and also its influence on the activation of non-targeted cells – that is, cells within the FOV not explicitly targeted with a beamlet. We performed 8 ~ 11 trials for each stimulation pattern. Cells not responding to photostimulation under any condition were excluded in this analysis (see Materials and Methods). We characterize the response rate at the individual cell (*Figure 3F*) and the ensemble level (*Figure 3G*). The former characterizes the response rate of individual targeted cells in any stimulation pattern, and the latter characterizes the percentage of responsive cells within a targeted ensemble (defined here as ensemble response rate). As the simultenously stimulated neurons number *M* increased, the response rate for both individual cells and ensembles remained high (both is 82 ± 9%, over all seven stimulation conditions). Although we had high targeting accuracy and reliability for exciting targeted cells, we also observed occasional activity in non-targeted cells (nonspecific activation) during photostimulation (*Figure 3H*). This was distance-dependent, and as the distance *d* between the non-targeted cells and their nearest targeted cells decreased, their probability of activation increased (*Figure 3H*). And, for the same *d*, this probability increased with *M*. The activation of the non-targeted cells may occur through different mechanisms, such as by direct stimulation (depolarization) of the cells through their neurites that course through the photostimulation region, or through synaptic activation by targeted cells, or by a combination of the two. In these experiments, we specifically used extremely long stimulation durations (480 ~ 962 ms) to maximally emulate an undesirable photostimulation scenario. The nonspecific activation was confined (half response rate) within *d* < 25 µm in all conditions (*M* = 3 ~ 27 across three planes spanning a volume of 240 × 240 × 100 µm$^3$). Nonspecific activation could be reduced by increasing excitation *NA* (which is currently limited by the relatively small size of the activation galvanometer mirrors), using somatic-restricted expression (*Pégard et al., 2017*; *Baker et al., 2016*; *Shemesh et al., 2017*), as well as sparse expression.

We then aimed to modulate relatively large groups of neurons in 3D. With the low-repetition-rate laser and hybrid scanning strategy (Discussion), the laser beam can be heavily spatially multiplexed to address a large amount of cells while maintaining a low average power. We performed photostimulation of 83 cells across an imaged volume of 480 × 480×150 µm$^3$ in layer 2/3 of V1 in awake mice (*Figure 4*). With a total power of 300 mW and an activation time of ~ 95 ms, we were able to activate more than 50 cells. In one experiment, we further sorted target cells into two groups (40 and 43 cells respectively) and photostimulated them separately. More than 30 cells in each group were successfully activated simultaneously with clear evoked calcium transient. In another example, more than 35 cells out of a target group of 50 cells responded (*Figure 4—figure supplement 1*). These large scale photostimulations (>=40 target cells; *Figure 4*), show that 78 ± 7% of cells in the target ensemble can be successfully activated (excluding cells that never respond in any of the tested photostimulation pattern, 8 ± 3%, see Materials and Methods). Nonspecific photoactivation

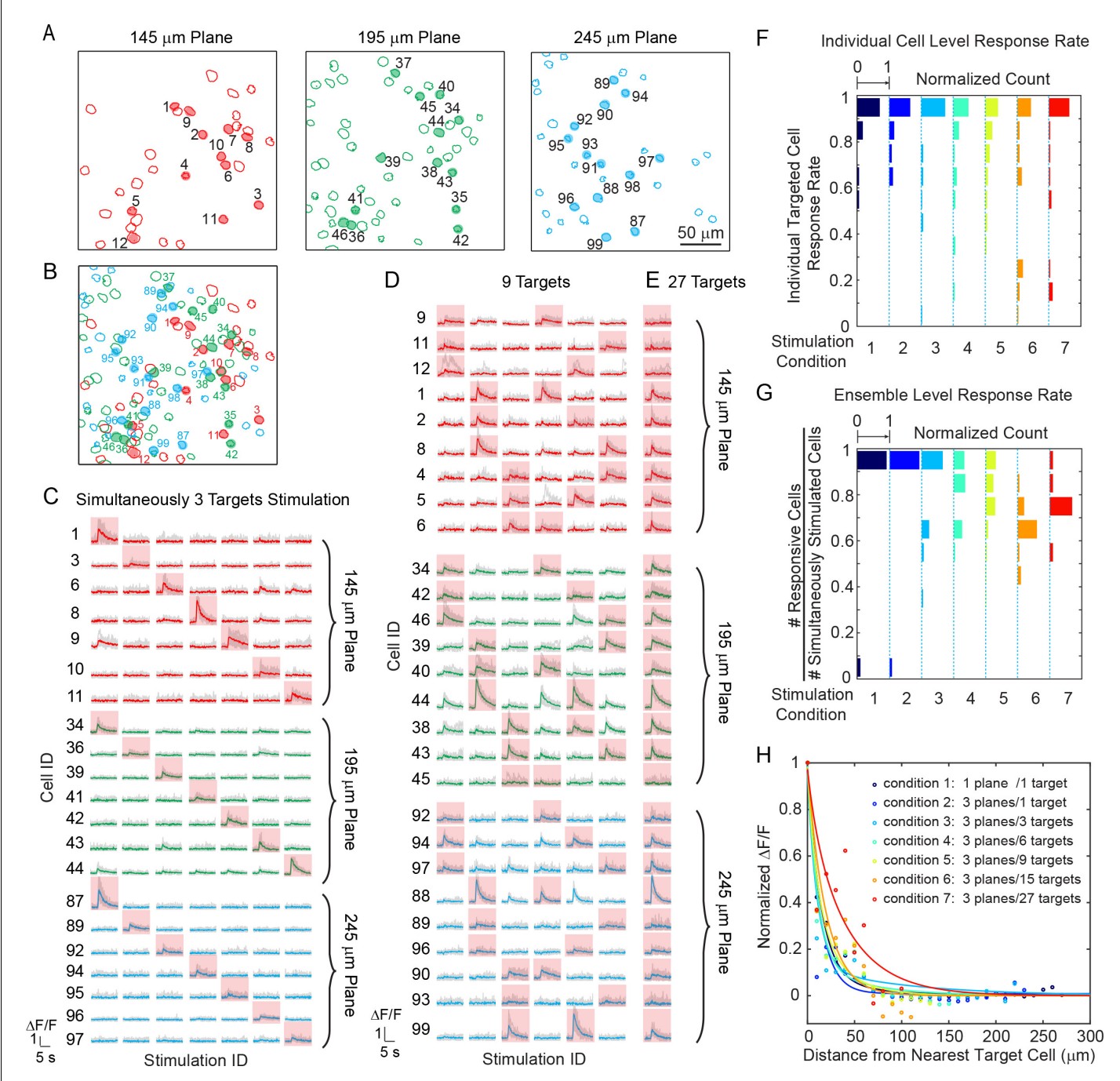

**Figure 3.** Simultaneous holographic photostimulation of pyramidal cells in vivo. (**A**) Contour maps showing the spatial location of the cells in three individual planes in mouse V1 (145 µm, 195 µm, and 245 µm from pial surface). Cells with shaded color are the targeted cells. (**B**) 2D overlap projection of the three planes in (**A**). (**C**)-(**E**) Representative photostimulation triggered calcium response of the targeted cells (indicated with red shaded background) and non-targeted cells, for different stimulation patterns. A total number of (**C**) 3, (**D**) 9, and (**E**) 27 cells across three planes were simultaneously photostimulated. The average response traces are plotted over those from the individual trials. (**F**) Histogram of individual targeted cell response rate (averaged across trials) in different stimulation conditions. The stimulation conditions are listed in (**H**). (**G**) Histogram of the percentage of responsive cells in a targeted ensemble across all trials in different stimulation conditions. (**H**) Response of the non-targeted cells to the photostimulation versus distance to their nearest targeted cell. ΔF/F is normalized to the averaged response of the targeted cells. The total number of photostimulation patterns for condition 1 ~ 7 in (**F~H**) is 34, 26, 12, 8, 6, 2, 1 respectively; and the total trial for each condition is 8 ~ 11. The mice were transfected with GCaMP6s and C1V1-mCherry. The photostimulation power is 4 ~ 5 mW/cell, and duration was 870 ms, 962 ms, and 480 ms for conditions 1, 2, and 3 ~ 7 respectively.

*Figure 3 continued on next page*

*Figure 3 continued*

DOI: https://doi.org/10.7554/eLife.32671.010

The following figure supplement is available for figure 3:

**Figure supplement 1.** Sequential photostimulation of individual pyramidal cells in layer 2/3 from mouse V1 in vivo.

DOI: https://doi.org/10.7554/eLife.32671.011

was more frequent for cells surrounded by target cells, but overall it was confined within 20 μm from the nearest target cell (*Figure 4F*). We also noted that cells that could be photoactivated individually or in a small ensemble may not get photoactivated when the number of target neurons increases. We hypothesize that this could be due to feed forward inhibition, as targeted pyramidal neurons may activate local interneurons, which then could suppress the firing of neighboring cells. These network interactions will be the subject of future study.

Nonspecific excitation can be minimized with sparse stimulation, by simply reducing the likelihood of stimulating directly adjacent cells. One naturally sparse pool of cells are cortical interneurons. Different interneuron classes participate in cortical microcircuits that could serve as gateways for information processing (*Muñoz et al., 2017*; *Karnani et al., 2014*). These interneurons are located sparsely in the cortex, yet are highly connected to excitatory populations (*Fino and Yuste, 2011*), and are known to strongly modulate cortical activity (*Tsumoto et al., 1979*). However, the effect of simultaneous stimulation of selective subset of interneurons with single cell resolution has not been studied in detail, as previous reports have largely relied on one-photon optogenetics where widespread activation is the norm (*Lee et al., 2012*; *Wilson et al., 2012*) [but see Ref. (*Karnani et al., 2016*) for single cell interneuron stimulations]. To explore this, we used our all-optical approach to examine the effect of photoactivating specific sets of interneurons in 3D on the activity of pyramidal cells that responded to visual stimuli in awake head-fixed mice (*Figure 5*).

Using viral vectors, we expressed Cre-dependent C1V1 in somatostatin (SOM) inhibitory interneurons (SOM-Cre mice), while simultaneously also expressing GCaMP6s in both pyramidal cells and interneurons, in layer 2/3 of mouse V1. We first imaged the responses of pyramidal cells across three planes (separated by ~ 45 μm each) to orthogonal visual stimuli consisting of drifting grating without photostimulation. We then simultaneously photostimulated a group of SOM cells ($M = 9$, with seven showing responses) across these three planes concurrently with the visual stimuli (*Figure 5A–C*; Materials and Methods). We observed a significant suppression ($p<0.05$, two-sample t-test) in response among 46% and 35% of the pyramidal cells that originally responded strongly to the horizontal and vertical drifting-grating respectively (*Figure 5A–E*). Moreover, the orientation selectivity of highly selective cells was largely abolished by SOM cell photoactivation (*Figure 5E*). This is consistent with reports that SOM cells inhibit nearby pyramidal cells with one-photon optogenetics *in vivo* (*Lee et al., 2012*; *Wilson et al., 2012*) or with two-photon glutamate uncaging *in vitro* (*Fino and Yuste, 2011*). Our two-photon approach provides high precision 3D manipulation over groups of cells (*Figure 5D*), and simultaneous readout of neuronal activity across the network in vivo. Thus, our approach could be useful for dissecting the excitatory and inhibitory interactions in cortical circuits in vivo.

## Discussion

We describe here a 3D all-optical method that could be used to map the functional connectivity of neural circuits and probe the causal relationships between the activity of neuronal ensembles and behavior. We extend previous in vivo methods from planar to volumetric targeting, and increase the total number of cells that could be simultaneously photoactivated. This represents a significant advance of precision optogenetics towards large spatial scales and volumes. The dual beam path microscope facilitates independent control of imaging and photostimulation lasers, and is thus well suited for controlling and detecting neural activity, without any disturbing or slow movements of the objective. In the following sections we discuss our rationale for the design of the microscope and evaluate the results.

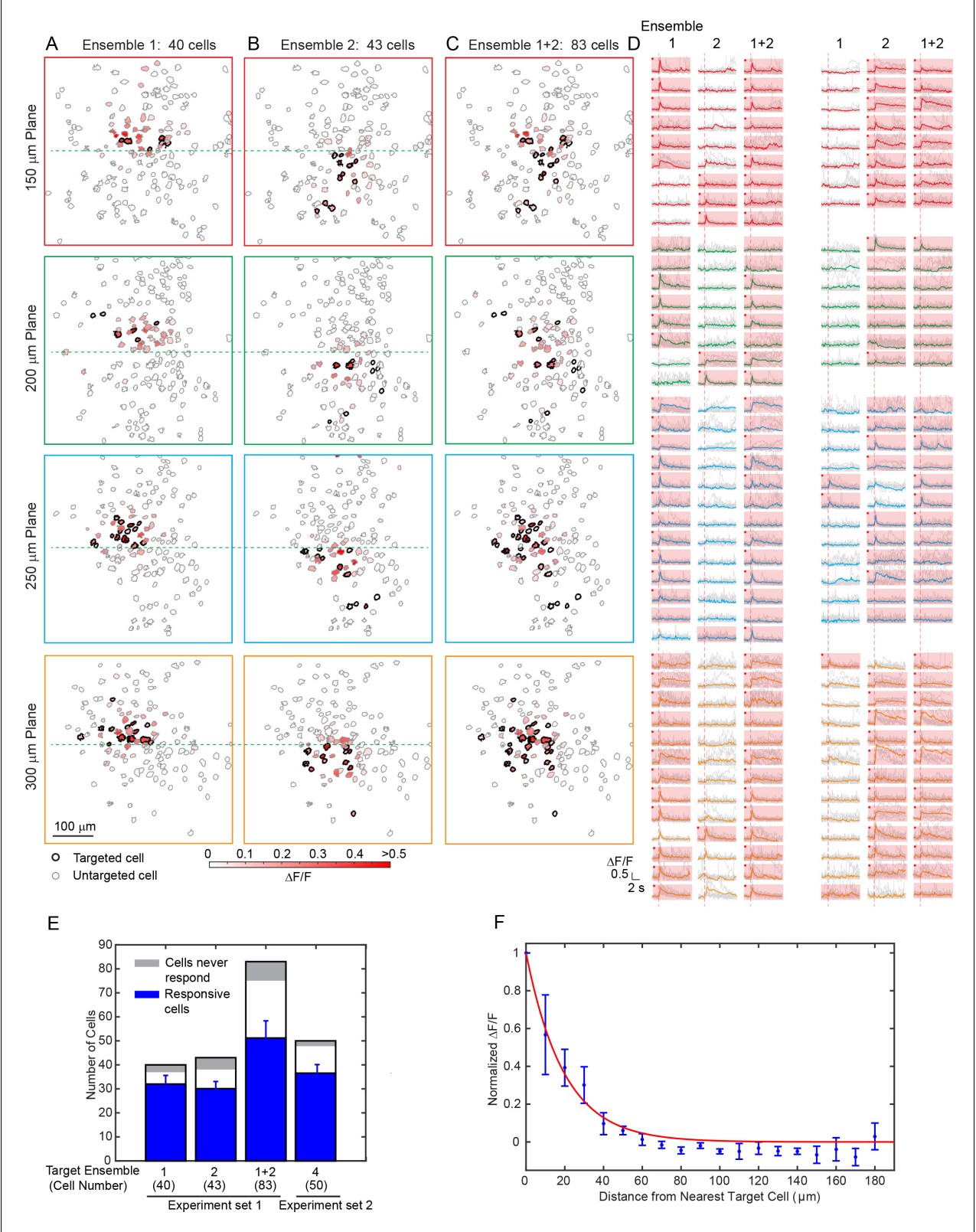

**Figure 4.** Large scale photostimulation of pyramidal cells in layer 2/3 of V1 in awake mice. (**A~C**) Simultaneous photostimulation of 40 cells, 43 cells and 83 cells across four planes in mouse V1 (150 μm, 200 μm, 250 μm and 300 μm from pial surface, with an imaged FOV of 480 × 480 μm² in each plane.). The contour maps show the spatial location of the cells in individual planes. Cells with black contour are the simultaneously targeted cells. The red shaded color shows the evoked ΔF/F in average. (**D**) Photostimulation triggered calcium response of the targeted cells (indicated with red shaded

*Figure 4 continued on next page*

*Figure 4 continued*

background) and non-targeted cells, corresponding to conditions shown in (**A**~**C**). The average response traces are plotted over those from a total of 11 individual trials. Those with a red dot indicate cells showing clear evoked calcium transient through manual inspection. (**E**) Number of target cells, number of total responsive cells across all trials, and cells that did not show any response in any photostimulation pattern, for four different photostimulation conditions. Condition 1 ~ 3 correspond to those in (**A**~**C**). Error bars are standard deviation over trials. (**F**) Response of the non-targeted cells to the photostimulation versus distance to their nearest targeted cell (for conditions shown in E). ΔF/F is normalized to the averaged response of the targeted cells. Error bars are standard error of the mean over different photostimulation conditions in (E). The mice were transfected with GCaMP6f and C1V1-mCherry. The photostimulation power was 3.6 ~ 4.8 mW/cell, and the duration was 94 ms (composing of 5 continuously repeated spiral scans).

DOI: https://doi.org/10.7554/eLife.32671.012

The following figure supplement is available for figure 4:

**Figure supplement 1.** Simultaneous photostimulation of 50 pyramidal cells in layer 2/3 of V1 in awake mice.

DOI: https://doi.org/10.7554/eLife.32671.013

## A- Minimization of laser power

To simultaneously photostimulate multiple cells with two-photon excitation, it is becoming common to use holographic approaches (*Nikolenko et al., 2008*; *Packer et al., 2012*; *Packer et al., 2015*; *Dal Maschio et al., 2017*; *Mardinly et al., 2017*; *Pégard et al., 2017*). Spatial light modulators can generate an 'arbitrary' 3D pattern on the sample, limited only by Maxwell's equations, and the space-bandwidth product of the modulation device. With SLMs, one can independently target a very large number of sites, far in excess of what we demonstrate here, but the number of addressable neurons is limited by the allowable power budget. Moreover, special care has to be taken to minimize the total power deposited on the brain, and avoid direct and indirect thermal effects (*Podgorski and Ranganathan, 2016*). We addressed this issue by using a hybrid holographic strategy and a low-repetition-rate laser for photostimulation, with high peak intensities for efficient two-photon excitation, but moderate average power. This allowed us to target a large group of cells with low average power (e.g. 83 targeted cells across an imaged volume of $480 \times 480 \times 150$ μm$^3$ in awake mice V1 layer 2/3 with 300 mW in total, *Figure 4*). As these cells generally are not targeted continuously, we do not expect any heat induced effects on cell health under our stimulation conditions (*Podgorski and Ranganathan, 2016*).

In our hybrid strategy, a group of beamlets is generated by the SLM that target the centroids of the desired neurons. Each discrete focal point in the hologram maintains sufficient axial confinement for typical inter-cell spacing. These beamlets are then rapidly spirally scanned over the neurons' cell bodies by post-SLM galvanometers. Several alternative scanless approaches exist: pure 3D holograms and another method combining holographic patterning and temporal focusing. The former approach directly generates the full 3D hologram covering the cell bodies of targeted neurons all at once (*Dal Maschio et al., 2017*). Though simplest, the full 3D hologram has a decreased axial resolution as its lateral extend increases (*Papagiakoumou et al., 2008*), and is subject to light contamination to the non-targeted cells, particularly in scattering tissues such as the mammalian brain. In contrast, temporal focusing (*Oron et al., 2005*; *Zhu et al., 2005*) decouples axial from lateral extent of the hologram by coupling the holographic pattern to a grating (*Papagiakoumou et al., 2008*) such that only one axial position in the sample has sufficient spectral content to generate a short laser pulse, thus tightening the axial confinement. Recent reports have extended this method to 3D stimulation(*Hernandez et al., 2016*;*Pégard et al., 2017*;*Accanto et al., 2017*). Regardless of the exact implementation, these scanless approaches require higher laser powers per cell in general than our hybrid method. For example, with typical photostimulation duration ($\geq$5 ms), about twice of the power is required using pure hologram compared with our hybrid strategy to achieve similar response in the same cells (*Figure 2*, *Figure 2—figure supplement 1*). It would likely require even *more* power for the same excitation with temporal focusing, as its tighter axial confinement would excite less of the membrane. On the other hand, the area-activation of scanless activation generally gives lower latencies and less jitter, compared to scanning strategies. However, as we show in our hybrid scanning approach, even with low powers and longer scan times, we can obtain latencies under 10 ms, with little jitter (<1 ms, *Figure 1—figure supplement 2*). Taken together, the spiral scan strategy we adapted requires a lower laser power budget per cell, and is very scalable towards

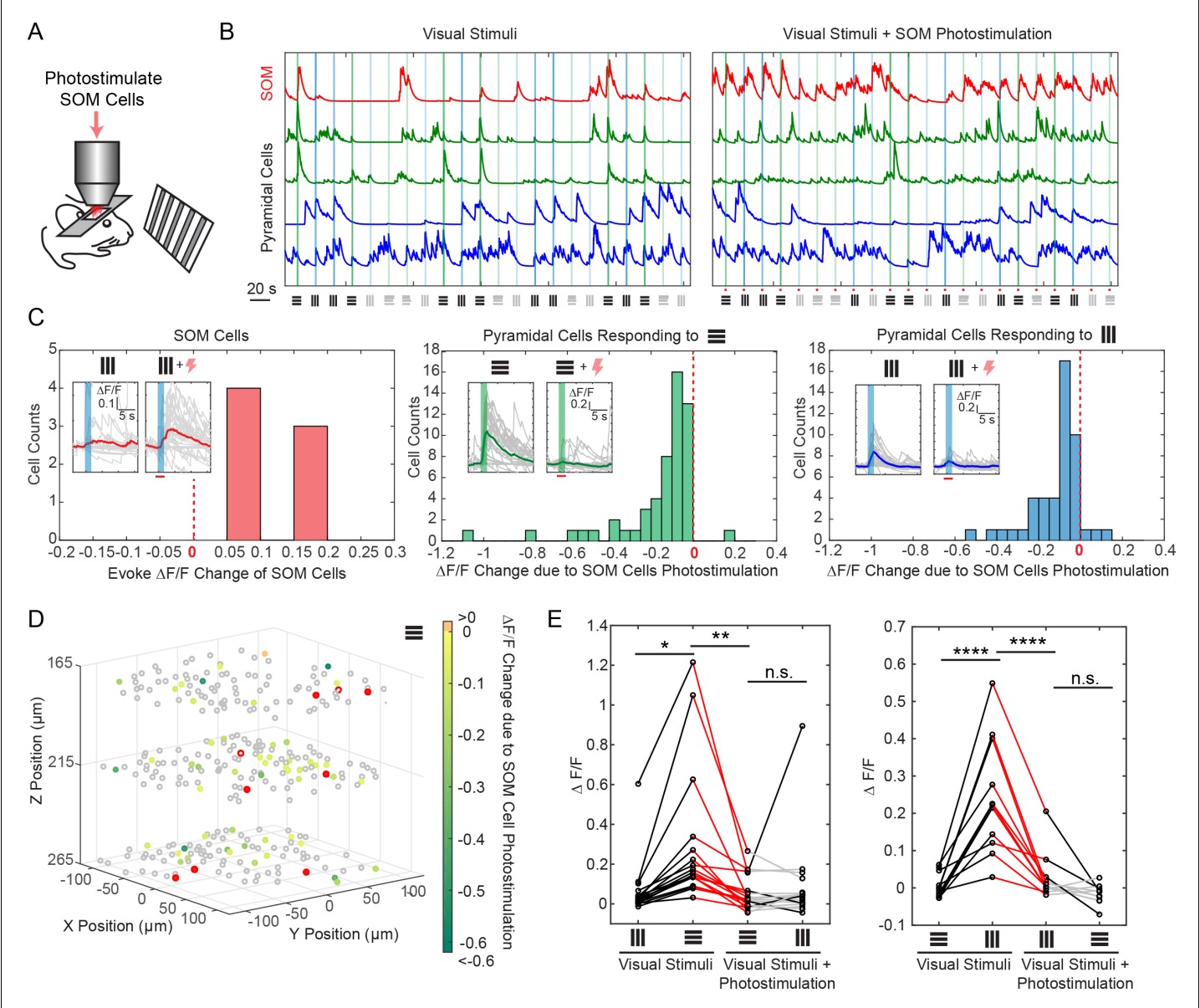

**Figure 5.** Selective photostimulation of SOM interneurons suppresses visual response of pyramidal cells in awake mice. (A) Experiment paradigm where the SOM cells were photostimulated when the mouse received drifting grating visual stimulation. (B) Normalized calcium traces (ΔF/F) of representative targeted SOM cells and pyramidal cells that are responding to visual stimuli, without (left panel) and with (right panel) SOM cell photostimulation. The normalization factor of the ΔF/F trace for each cell stays the same across the two conditions. The shaded regions indicate the visual stimuli period. The symbols at the bottom of the graph indicate the orientations and contrast of the drifting grating (black, 100% contrast; gray, 10% contrast).
(C) Histogram of the visual stimuli evoked ΔF/F change for different cell populations that show significant activity change (p<0.05, two-sample t-test over ~ 30 trials) due to SOM cell photostimulation (M = 9, simultaneously photostimulated). Left panel, targeted SOM cells (7 out of 9 show significant responses to photostimulation). Middle and right panels, pyramidal cells responding to horizontal or vertical drifting-gratings respectively. The inset compares the activity of a representative cell without and with targeted SOM cell photostimulation; the shaded regions indicate the visual stimuli period; the red bar indicates the photostimulation period. (D) Spatial map of all recorded cells. Pyramidal cells responding to horizontal drifting-gratings and showing significant visual stimuli evoked ΔF/F change due to SOM cell photostimulation (p<0.05, two-sample t-test over ~30 trials) [cell population in the middle panel of C] are color coded according to their ΔF/F change. The targeted SOM cells are outlined in red, and those responding are shaded in red. (E) Comparison of the orientation selectivity in normal situation and with SOM cells photostimulation, for a cell population that normally have strong orientation selectivity but responsive to SOM cells photostimulation. During SOM cell photostimulation, their selectivity is largely abolished (one-way ANOVA test). For individual cells, black and red lines indicate a significant difference in the visual stimuli evoked ΔF/F between the two conditions that the lines connect with (~30 trials, p<0.05, two-sample t-test), whereas gray lines indicate no significant

*Figure 5 continued*

difference. The SOM-cre mice were transfected with GCaMP6s and Cre-dependent C1V1-mCherry. The duration of visual stimuli was 2 s. The photostimulation power was ~ 6 mW/cell, and the duration was 2.8 s (composing of 175 continuously repeated spiral scans).

DOI: https://doi.org/10.7554/eLife.32671.014

activating large number of simultaneously targeted cells, making it a practical tool to study ensembles in neural circuits.

One key strategy we exploited to lower the total average laser power in patterned photostimulation was to employ a low-repetition-rate laser for photostimulation. The average laser power $P_{ave}$ scales with the product of laser peak power $P_{peak}$ and pulse repetition rate $f_{rep}$. Since the laser beam is split into $M$ beamlets to target $M$ individual cell, the two-photon excitation for each cell scales with $(P_{peak}/M)^2$ (*Denk et al., 1990*). To maintain the required $P_{peak}$ for a large $M$, we reduced $f_{rep}$ instead of increasing $P_{ave}$. The two-photon photostimulation laser we used had a low $f_{rep}$ (200 kHz ~ 1 MHz), leading to a significant increase in $P_{peak}$ and thus the number of possible simultaneously targeted cells $M$, with the same $P_{ave}$. We note that most opsins open ion channels, the average open time is much longer than the laser's interpulse interval ($1/f_{rep}$), and multiple ions can be conducted during each photostimulation. This is in contrast to fluorescence, where at most a single photon is emitted for each absorption, and the lifetime is significantly shorter than the interpulse interval. Thus opsins are ideal targets for low-repetition rate, high peak power excitation. In addition, the repetition rate should be balanced with the photostimulation duration. When the photostimulation duration is very short (e.g. 1 ms), the whole cell body might not be covered well with enough pulses in the spiral scan approach. In these scenarios, a higher repetition rate could be more favorable. The optimal conditions will likely be cell- and opsin-dependent, but would be expected to follow our trends.

## B- Volumetric Imaging

We choose an ETL for volumetric imaging, because of its low cost and good performance for focusing. Many other options exist including SLM (*Yang et al., 2016*), ultrasound lens (*Kong et al., 2015*), remote focusing (*Botcherby et al., 2012*; *Sofroniew et al., 2016*) and acousto optic deflector (*Duemani Reddy et al., 2008*; *Grewe et al., 2010*; *Katona et al., 2012*); see Ref. (*Yang and Yuste, 2017*), for a complete review. One future modification could be replacing the ETL with a second SLM to perform multiplane imaging (*Yang et al., 2016*) and adaptive optics (*Ji, 2017*), which could increase the frame rate and improve the imaging quality.

## C- Minimizing Cross-talk between Imaging and Photostimulation

Another important consideration in our all-optical method was to minimize the cross-talk between imaging and photostimulation. We chose the calcium indicator GCaMP6 and the red-shift opsin C1V1-mCherry, which has a minimized excitation spectrum overlap. Nevertheless, there is still a small cross-talk between the two, as C1V1 has a blue absorption shoulder, and GCaMP6 has a red shifted absorption tail. The first cross-talk affects neuronal excitability, and is the result of photostimulation by the imaging laser. Although the C1V1 we used was red-shifted, it can still be excited at 920 ~ 940 nm, the typical wavelengths used to image GCaMP6. This cross-talk highly depends on the relative expression of the calcium indicators and opsin (*Rickgauer et al., 2014*; *Packer et al., 2015*). For this reason, the imaging laser power was kept as low as possible to values that are just sufficient for imaging. But if the calcium indicator is weakly expressed, hence naturally dim, the increased imaging power may bias the neuronal excitability. Indeed, our cell-attached electrophysiology recording indicates that neuron firing rate has a trend to increase as the imaging laser power increases. However, we found no significant difference of the firing rate under our normal volumetric imaging conditions (*Figure 1—figure supplement 3*), where the laser power was typically below 50 mW and could be up to 80 mW for layers deeper than ~ 250 µm. Nevertheless, as red indicators keep improving, a future switch toward 'blue' opsins again will be desirable to reduce the spectral overlap between opsin and indicator.

The second type of cross-talk affects the high fidelity recording of neural activity, and is caused by fluorescence (or other interference) generated by the photostimulation laser directly, which may

cause background artifact on the calcium signal recording. To avoid this, in our experiments we use a narrow filter (passband: 500 nm ~ 520 nm) for GCaMP6 signal detection. C1V1 is co-expressed with mCherry, which has negligible fluorescence at the filter's passband. But, in addition, GCaMP6 can still be excited at the photostimulation laser's wavelength at 1040 nm. Typically this fluorescence is weak and does not impact the data analysis (e.g. *Figure 3*). However, if the baseline of GCaMP6 is relatively high or the number of simultaneously targeted neurons is large, it could cause a significant background artifact in the calcium imaging, identified as sharp rise and then sharp decay of fluorescence signals (*Figure 1—figure supplement 4*). If the photostimulation duration is short (e.g. *Figure 4*, only one frame appears to have the artifact), and stimulation frequency infrequent, the impacted frames could simply be deleted with negligible data loss. But if the photostimulation duration is long (e.g. *Figure 5*), the calcium imaging movie can be pre-processed so that the mesh grid shape background is replaced by their adjacent pixel value (see Materials and Methods). The 'mesh' arises because the interpulse interval of the laser is greater than the pixel rate, so only selected pixels are compromised. The grid is non-uniform in the image because of the non-uniform resonant scanner speed. This pre-processing significantly suppresses the artifacts while maintaining the original signal. Nevertheless, to avoid any analysis bias, the neuronal response can be further approximated by measuring the ΔF/F signal right after the photostimulation, when there is no background artifact. Also, an alternative method is to gate the PMT, or the PMTs output during the photostimulation pulse, thought this requires dedicated additional electronics. In this case, there will be 'lost' signal, and this can be treated similarly by filling in the data with interpolation. Finally, the constrained nonnegative matrix factorization algorithm (*Pnevmatikakis et al., 2016*) used to extract the fluorescence signal could also help, as it can identify the photostimulation artifact as part of the background and subtract it from the signal. With these corrections, the photostimulation artifacts can be eliminated from the extracted fluorescence trace in *Figure 3*~5.

## D- Nonspecific Activation

One strategy to reduce nonspecific stimulation is to reduce the size of the PSF by increasing the NA. In our current set of experiments, we use a relatively low excitation NA (~0.35) beam that is limited by the small mirror size (3 mm) of the post-SLM galvanometric scanners. Increasing the mirror size is a straightforward future improvement that would increase this NA, and decrease the axial point spread function. This would also improve the effective axial resolution of photostimulation (currently ~ 20 μm, measured by displacing the 12 μm diameter spiral pattern relative to the targeted neuron, *Figure 1—figure supplement 2*), and thus reduce the nonspecific activation of the non-targeted cells. Another approach to reduce the nonspecific activation is to use a somatic-restricted opsin. Somatic-restricted opsins were reported recently (*Baker et al., 2016*; *Pégard et al., 2017*; *Shemesh et al., 2017*), and showed reduced, but not eliminated, activation of non-targeted cells in vitro. Finally, it remains possible that a significant number of nonspecific activated cells occur through physiological synaptic activation by the photostimulated neurons.

## Future outlook

Our method could have wide utility in neuroscience. We demonstrate the successful manipulation of the targeted neural microcircuits in awake head-fixed behaving mice by photostimulating a targeted group of interneurons (*Figure 5*), and we expect this 3D all-optical method would find its many other applications in dissecting the neural circuits. Though we only targeted neurons in cortical layers 2/3, the total targetable range of the SLM can be more than 500 μm (*Yang et al., 2016*), thus covering layers 2/3 and 5 simultaneously. Questions such as how neural ensembles are being organized across different cortical layers, and how different neural assemblies across a 3D volume interact with each other can now be directly explored. Indeed, by identifying behavior-related neural ensemble using closed-loop optogenetics (*Grosenick et al., 2015*; *Carrillo-Reid et al., 2017*), one may be able to precisely control the animal behavior, which could have a significant impact in attempts to decipher neural codes and also provide an optical method for potential treatment of neurological and mental diseases in human subjects.

## Materials and methods

### Microscope design

The optical setup is illustrated in *Figure 1A*, which is composed of two femtosecond pulse lasers and a custom-modified two-photon laser scanning microscope (Ultima In Vivo, Bruker Corporation, Billerica, Massachusetts). The laser source for imaging is a pulsed Ti:sapphire laser (Chameleon Ultra II, Coherent, Inc., Santa Clara, California). Its wavelength is tuned to 940 nm for GCaMP6s or GCaMP6f imaging or 750 nm for mCherry imaging respectively. The laser power is controlled with a Pockels cell (350–160-BK Pockels cell, 302RM controller, Conoptics, Inc., Danbury, Connecticut). The laser beam is expanded by a 1:3.2 telescopes (f = 125 mm and f = 400 mm) and coupled to an ETL (EL-10-30-C-NIR-LD-MV, Optotune AG, Dietikon, Switzerland) with a clear aperture of 10 mm in diameter. The transmitted beam is rescaled by a 3.2:1 telescope (f = 400 mm and f = 125 mm) and imaged onto a resonant scanner and galvanometric mirror, both located at the conjugate planes to the microscope's objective pupil. The beam is further scaled by a 1:1.33 telescope before coupled into a scan lens (f = 75 mm), a tube lens (f = 180 mm) and the objective lens (25x N.A. 1.05 XLPlan N, Olympus Corporation, Tokyo, Japan), yielding an excitation NA ~ 0.45. The laser can also be directed to a non-resonant scanning path (without ETL) where both X and Y scanning are controlled by galvanometric mirrors. The fluorescence signal from the sample is collected through the objective lens and split at a dichroic mirror (HQ575dcxr, 575 nm long pass, Chroma Technology Corp., Bellows Falls, Vermont) to be detected in two bi-alkali photomultiplier tubes, one for each wavelength range. Two different bandpass filters (510/20–2P, and 607/45–2P, Chroma Technology Corp., Bellows Falls, Vermont) are placed in front of the corresponding PMT respectively.

The optical path for the photostimulation is largely independent from the imaging, except that they combine at a dichroic mirror (T1030SP, 1030 nm short pass, Chroma Technology Corp., Bellows Falls, Vermont) just before the scan lens, and then share the same optical path. The laser source for photostimulation is a low repetition rate (200 kHz ~ 1 MHz) pulse-amplified laser (Spirit 1040–8, Spectra-physics, Santa Clara, California), operating at 1040 nm wavelength. Its power is controlled by a Pockels cell (1147-4-1064 Pockels cell, 8025RS-H-2KV controller, FastPulse Technology, Saddle Brook, New Jersey). A $\lambda/2$ waveplate (AHWP05M-980, Thorlabs, Inc. Newton, New Jersey) is used to rotate the laser polarization so that it is parallel to the active axis of the spatial light modulator (HSP512-1064, $7.68 \times 7.68$ mm$^2$ active area, $512 \times 512$ pixels; Meadowlark Optics, Frederick, Colorado). The beam is expanded by two telescopes (1:1.75, f = 100 mm and f = 175 mm; 1:4, f = 50 mm and f = 200 mm) to fill the active area of the SLM. The reflected beam from the SLM is scaled by a 3:1 telescope (f = 300 mm and f = 100 mm) and imaged onto a set of close-coupled galvanometer mirrors, located at the conjugate plane to the microscope's objective pupil. A beam block made of a small metallic mask on a thin pellicle is placed at the intermediate plane of this telescope to remove the zeroth-order beam. The photostimulation laser beam reflected from the galvanometer mirrors are then combined with the imaging laser beam at the 1030 nm short pass dichroic mirror.

The imaging and photostimulation is controlled by a combination of PrairieView (Bruker Corporation, Billerica, Massachusetts) and custom software (*Yang, 2018*) running under MATLAB (The Mathworks, Inc. Natick, Massachusetts) on a separate computer. The Matlab program was developed to control the ETL through a data acquisition card (PCIe-6341, National Instrument, Austin, Texas) for volumetric imaging, and the SLM through PCIe interface (Meadowlark Optics, Frederick, Colorado) for holographic photostimulation (*Yang, 2018*). The two computers are synchronized with shared triggers. At the end of each imaging frame, a signal is received to trigger the change of the drive current (which is converted from a voltage signal from the data acquisition card by a voltage-current converter [LEDD1B, Thorlabs, Inc. Newton, New Jersey]) of the ETL, so the imaging depth is changed for the following frame. The range of the focal length change on sample is ~+90 μm ~ −200 μm ('+" means longer focal length). The intrinsic imaging frame rate is ~ 30 fps with $512 \times 512$ pixel image. The effective frame rate is lower as we typically wait 10 ~ 17 ms in between frames to let the ETL fully settle down at the new focal length. The control voltage of the Pockels cell is switched between different imaging planes to maintain image brightness. The typical imaging power is < 50 mW, and could be up to 80 mW for layers deeper than ~ 250 μm.

The Matlab programs to control the ETL for volumetric imaging and SLM for holographic photostimulation (*Yang, 2018*) is available at https://github.com/wjyangGithub/Holographic-

Photostimulation-System with a GNU General Public License, version 3 (copy archived at https://github.com/elifesciences-publications/Holographic-Photostimulation-System).

## SLM hologram and characterization

The phase hologram on the SLM, $\phi(u,v)$, can be expressed as:

$$\phi(u,v) = phase\left\{ \sum_{i=1}^{M} A_i e^{2\pi j \left\{ x_i u + y_i v + \left[ Z_2^0(u,v) C_2^0(z_i) + Z_4^0(u,v) C_4^0(z_i) + Z_6^0(u,v) C_6^0(z_i) \right] \right\}} \right\} \tag{1}$$

where $[x_i, y_i, z_i]$ $(i = 1,2\ldots M)$ is the coordinate of the cell body centroid ($M$ targeted cells in total), and $A_i$ is the electrical field weighting coefficient for the $i^{th}$ target (which controls the laser power it receives). $Z_m^0(u,v)$ and $C_m^0(z_i)$ are the Zernike polynomials and Zernike coefficients, respectively, which sets the defocusing and compensates the first-order and second-order spherical aberration due to defocusing. Their expressions are shown in *Table 1*. The hologram can also be generated by 3D Gerchberg-Saxton algorithm, with additional steps to incorporate spherical aberration compensation. We adapt *Equation (1)* as a simpler method. For the experiments in *Figure 2*, and *Figure 2—figure supplement 1*, the Gerchberg-Saxton algorithm is used to generate a disk with a diameter similar to the neurons.

To match the defocusing length set in SLM with the actual defocusing length, we adjusted the 'effective N.A.' in the Zernike coefficients following the calibration procedure described in Ref. (*Yang et al., 2016*). To register the photostimulation beam's targeting coordinate in lateral directions, we projected 2D holographic patterns to burn spots on the surface of an autofluorescent plastic slide and visualized them by the imaging laser. An affine transformation can be extracted to map the coordinates. We repeated this registration for every 25 µm defocusing depth on the sample, and applied a linear interpolation to the depths in between. An alternative method to register the targeting coordinate is to set the photostimulation laser in imaging mode, actuate the SLM for different lateral deflection, and extract the transform matrix from the acquired images and that acquired from the imaging laser. To characterize the lateral registration error, we actuated the SLM and burned spots on the surface of an autofluorescent plastic slide across a field of view of 240 µm x 240 µm with a 7 × 7 grid pattern. We then imaged the spots pattern with the imaging laser and measured the registration error. This was repeated for different SLM focal depths. To characterize the axial registration error, we used the photostimulation laser to image a slide with quantum dots sample. The SLM was set at different focal depths, and a z-stack was acquired for each setting to measure the actual defocus and thus the axial registration error. In all these registration and characterization procedures, we used water as the media between the objective and the sample, and we kept the focus of the photostimulation laser at the sample surface by translating the microscope stage axially. We note that the refractive index of the brain tissue is slightly different from that of

**Table 1.** Expression of Zernike polynomials and Zernike coefficients in *Equation (1)*.

**Defocus**

| | |
|---|---|
| Zernike polynomials | $Z_2^0(u,v) = \sqrt{3}[2(u^2 + v^2) - 1]$ |
| Zernike coefficients | $C_2^0(z) = \frac{nkz\sin^2\alpha}{8\pi\sqrt{3}} \left( 1 + \frac{1}{4}\sin^2\alpha + \frac{9}{80}\sin^4\alpha + \frac{1}{16}\sin^6\alpha + \cdots \right)$ |

**First-order spherical aberration**

| | |
|---|---|
| Zernike polynomials | $Z_4^0(u,v) = \sqrt{5}\left[ 6(u^2 + v^2)^2 - 6(u^2 + v^2) + 1 \right]$ |
| Zernike coefficients | $C_4^0(z) = \frac{nkz\sin^4\alpha}{96\pi\sqrt{5}} \left( 1 + \frac{3}{4}\sin^2\alpha + \frac{15}{18}\sin^4\alpha + \cdots \right)$ |

**Second-order spherical aberration**

| | |
|---|---|
| Zernike polynomials | $Z_6^0(u,v) = \sqrt{7}\left[ 20(u^2 + v^2)^3 - 30(u^2 + v^2)^2 + 12(u^2 + v^2) - 1 \right]$ |
| Zernike coefficients | $C_6^0(z) = \frac{nkz\sin^6\alpha}{640\pi\sqrt{7}} \left( 1 + \frac{5}{4}\sin^2\alpha + \cdots \right)$ |

$n$, refractive index of media between the objective and sample; $k$, the wavenumber; $z$, the axial shift of the focus plane in the sample; $u$, $v$, coordinates on the SLM phase mask; $n\sin\alpha$, the NA of the objective.

DOI: https://doi.org/10.7554/eLife.32671.015

water (~2%), and this could cause an axial shift of the calibration. This could be corrected in the Zernike coefficients. In practice, we found this effect is negligible, as the typical focal shift by the SLM is relatively small (<150 µm) and the axial PSF is large.

Due to the chromatic dispersion and finite pixel size of SLM, the SLM's beam steering efficiency drops with larger angle, leading to a lower beam power for targets further away from the center field of view (in xy), and nominal focus (in z). The characterization result is shown in *Figure 1—figure supplement 1*. A linear compensation can be applied in the weighting coefficient $A_i$ in *Eq. (1)* to counteract this non-uniformity. In practice, these weighting coefficients can be adjusted such that the targeted neurons show clear response towards photostimulation.

Before each set of experiments on animals, we verify the system (laser power, targeting accuracy, power uniformity among different beamlets from the hologram) by generating groups of random spots through holograms, burning the spots on an autofluorescent plastic slide, and comparing the resultant image with the desired target.

## Animals and surgery

All experimental procedures were carried out in accordance with animal protocols approved by Columbia University Institutional Animal Care and Use Committee. Multiple strains of mice were used in the experiment, including C57BL/6 wild-type and SOM-cre (*Sst*-cre) mice (stock no. 013044, The Jackson Laboratory, Bar Harbor, Maine) at the age of postnatal day (P) 45–150. Virus injection was performed to layer 2/3 of the left V1 of the mouse cortex, 3 ~ 12 weeks prior to the craniotomy surgery. For the C57BL/6 wild-type mice, virus AAV1-syn-GCaMP6s (or AAV1-syn-GCaMP6f) and AAVDJ-CaMKII-C1V1-(E162T)-TS-p2A-mCherry-WPRE was mixed and injected for calcium imaging and photostimulation; virus AAV8-CaMKII-C1V1-p2A-EYFP was injected for electrophysiology. For the SOM-cre (*Sst*-cre) mice, virus AAV1-syn-GCaMP6s and AAVDJ-EF1a-DIO-C1V1-(E162T)-p2A-mCherry-WPRE was mixed and injected. The virus was front-loaded into the beveled glass pipette (or metal pipette) and injected at a rate of 80 ~ 100 nl/min. The injection sites were at 2.5 mm lateral and 0.3 mm anterior from the lambda, putative monocular region at the left hemisphere.

After 3 ~ 12 weeks of expression, mice were anesthetized with isoflurane (2% by volume, in air for induction and 1–1.5% during surgery). Before surgery, dexamethasone sodium phosphate (2 mg per kg of body weight; to prevent cerebral edema) were administered subcutaneously, and enrofloxacin (4.47 mg per kg) and carprofen (5 mg per kg) were administered intraperitoneally. A circular craniotomy (2 mm in diameter) was made above the injection cite using a dental drill. A 3 mm circular glass coverslip (Warner instruments, LLC, Hamden, Connecticut) was placed and sealed using a cyanoacrylate adhesive. A titanium head plate with a 4 mm by 3.5 mm imaging well was attached to the skull using dental cement. After surgery, animals received carprofen injections for 2 days as post-operative pain medication. The imaging and photostimulation experiments were performed 1 ~ 21 days after the chronic window implantation. During imaging, the mouse is either anesthetized with isoflurane (1–1.5% by volume in air) with a 37°C warming plate underneath or awake and can move freely on a circular treadmill with its head fixed.

## Visual stimulation

Visual stimuli were generated using MATLAB and the Psychophysics Toolbox (*Brainard, 1997*) and displayed on a monitor (P1914Sf, 19-inch, 60 Hz refresh rate, Dell Inc., Round Rock, Texas) positioned 15 cm from the right eye, at 45° to the long axis of the animal. Each visual stimulus session consisted of four different trials, each trial with a 2 s drifting square grating (0.04 cycles per degree, two cycles per second), followed by 18 s of mean luminescence gray screen. Four conditions (combination of 10% or 100% grating contrast, 0° or 90° drifting grating direction) were presented in random order in the four trials in each session.

## Photostimulation parameters

The pulse repetition rate of the photostimulation laser used in the experiment is 500 kHz or 1 MHz. The photostimulation laser beam is split into multiple foci, and spirally scanned (~12 µm outer spiral diameter, 8 ~ 50 rotations with progressively shrinking radius; the whole spiral can be continuously repeatedly scanned) by a pair of post-SLM galvanometric mirror over the cell body of each target cell. For neurons in layer 2/3 of mice V1, the typical average power used for each spot is 2 mW ~ 5

mW. When studying the photostimulation effect on the non-targeted cells (*Figure 3*), we specifically used long photostimulation durations (480 ms ~ 962 ms) to emulate an undesirable photostimulation scenario. In the normal condition, the photostimulation duration is < 100 ms, which was composed of multiple continuously repeated spiral scans, each lasting < 20 ms (*Figure 4*). In the experiments where short photostimulation duration (≤20 ms, *Figure 2*) is used, the stimulation was composed of a single spiral scan which consists of ~ 50 rotations with progressively shrinking radius. In the experiment that the SOM cells were photostimulated when the mouse were receiving visual stimuli (*Figure 5*), the photostimulation started 0.5 s before the visual stimuli, and ended 0.3 s after the visual stimuli finished. Since the visual stimuli lasted for 2 s, the photostimulation lasted for 2.8 s. This long photostimulation was composed of 175 continuously repeated spiral scans, each lasting ~ 16 ms. In our experiments, the lateral separation of the simultaneously targeted cell ranges from ~ 10 μm to ~ 315 μm, and the axial separation ranges from 30 μm to 150 μm.

## Data analysis

The recording from each plane was first extracted from the raw imaging files, followed by motion correction using a pyramid approach (*Thévenaz et al., 1998*) or fast Fourier transform-based algorithm (*Dubbs et al., 2016*). A constrained nonnegative matrix factorization (CNMF) algorithm (*Pnevmatikakis et al., 2016*) was used to extract the fluorescence traces (ΔF/F) of the region of interested (i.e. neuron cell bodies in the field of view). The CNMF algorithm also outputs a temporally deconvolved signal, which is related to the firing event probability. The ΔF/F induced by the photostimulation was quantified with the mean fluorescence change during the photostimulation period over the mean fluorescence baseline within a 0.5 ~ 2 s window before the photostimulation.

To detect the activity events from each recorded neuron, we typically thresholded the temporally deconvolved ΔF/F signal with at least two standard derivations from the mean signal. Independently, a temporal first derivative is applied to the ΔF/F trace. The derivative is then threshold at least two standard derivations from the mean. At each time point, if both are larger than the threshold, an activity event is recorded in binary format. In case the auto-detected activity event has large deviations from manual inspection (based on typical shapes of calcium transient), the thresholding value is adjusted so that the overall auto-detection agrees with manual inspection.

A cell is determined as not responding to photostimulation if there is no single activity event detected or no typical action-potential-corresponding calcium transient during photostimulation period for multiple trials. These non-responding cells could be due to a poor expression of C1V1.

Any GCaMP can generate fluorescence background during photostimulation (Discussion). This background would reduce the sensitivity of the calcium imaging. Since the pixel rate (~8.2 MHz) of the calcium imaging recording is much faster than the photostimulation laser's pulse repetition rate (200 kHz ~ 1 MHz), the fluorescence background appears to be a mesh grid shape in the calcium imaging movie (*Figure 1—figure supplement 4*). Typically it is small and does not impact the above data analysis (e.g. *Figure 3*). In the case that it is strong, if the photostimulation duration is short (e.g. *Figure 4*, only one frame appears to have the artifact), the impacted frames can be deleted with negligible data loss. If the photostimulation duration is long (e.g. *Figure 5*), the recorded frames during photostimulation are pre-processed to suppress this background artifact (*Figure 1—figure supplement 4*). To detect the pixels having this artifact, we consider both their fluorescence value and their geometry. First we detect candidate pixels by identifying pixels whose value is significantly higher from the average value calculated from a few frames just before and just after the stimulation. Second, these candidate pixels are tested for connectedness within every horizontal and vertical line of each frame, and the width of the connections compared to that expected based on the stimulation condition. If both these conditions hold, these pixels are marked as 'contaminated' and the fluorescence value at these pixels during the stimulation are replaced by those in their adjacent 'clean' pixels. This pre-processing significantly suppresses the artifacts while maintaining the original signal. Nevertheless, to avoid any analysis bias, we further approximated the neuronal response by using the ΔF/F signal just after the photostimulation, when there was no background artifact. The same analysis procedure was implemented to the control experiment when there was no photostimulation.

The orientation selectivity index and preference of the visual stimuli is calculated as the amplitude and sign of $(\Delta F/F|_{90} - \Delta F/F|_0) / (\Delta F/F|_{90} + \Delta F/F|_0)$ respectively, where $\Delta F/F|_{90}$ and $\Delta F/F|_0$ is the mean ΔF/F during the visual stimuli with 90° and 0° drifting grating respectively.

## In vivo electrophysiological recordings

Mice were head-fixed and anaesthetized with isoflurane (1.5 ~ 2%) throughout the experiment. Dura was carefully removed in the access point of the recording pipette. 2% agarose gel in HEPES-based artificial cerebrospinal fluid (ACSF) (150 mM NaCl, 2.5 mM KCl, 10 mM HEPES, 2 mM $CaCl_2$, 1 mM $MgCl_2$, pH was 7.3) was added on top of the brain to avoid movement artifacts. Patch pipettes of 5 ~ 7 MΩ pulled with DMZ-Universal puller (Zeitz-Instrumente Vertriebs GmbH, Planegg, Germany) were filled with ACSF containing 25 μM Alexa 594 to visualize the tip of the pipettes. C1V1-expressing cells were targeted using two-photon microscopy in vivo. During recordings, the space between the objective and the brain was filled with ACSF. Cell-attached recordings were performed using Multiclamp 700B amplifier (Molecular Devices, Sunnyvale, California), in voltage-clamp mode. The sampling rate was 10 kHz, and the data was low-pass filtered at 4 kHz using Bessel filter.

## Acknowledgements

This work was supported by the NEI (DP1EY024503, R01EY011787, R21EY027592), NIMH (R01MH101218, R01MH100561, R41MH100895, R44MH109187), and DARPA contracts W91NF-14-1-0269 and N66001-15-C-4032. This material is based upon work supported by, or in part by, the U. S. Army Research Laboratory and the U.S. Army Research Office under contract number W911NF-12-1-0594 (MURI). W Yang holds a career award at the scientific interface from Burroughs Wellcome Fund. Y Bando holds a fellowship from Uehara Memorial Foundation. The authors thank Reka Letso, Mari Bando, and Azi Hamzehei for virus injection of the mice; Jae-eun Kang Miller for mice preparation; Sean Quirin for the initial software for SLM control; Adam Packer and Alan Mardinly for fruitful discussions.

## Additional information

### Competing interests

Darcy S Peterka, Rafael Yuste: is listed as an inventor of the following patent: "Devices, apparatus and method for providing photostimulation and imaging of structures" (United States Patent 9846313). The other authors declare that no competing interests exist.

### Funding

| Funder | Grant reference number | Author |
|---|---|---|
| Burroughs Wellcome Fund | 1015761 | Weijian Yang |
| Uehara Memorial Foundation | | Yuki Bando |
| National Eye Institute | R21EY027592 | Darcy S Peterka |
| National Institute of Mental Health | R44MH109187 | Darcy S Peterka |
| National Eye Institute | DP1EY024503 | Rafael Yuste |
| National Eye Institute | R01EY011787 | Rafael Yuste |
| National Institute of Mental Health | R01MH101218 | Rafael Yuste |
| National Institute of Mental Health | R01MH100561 | Rafael Yuste |
| National Institute of Mental Health | R41MH100895 | Rafael Yuste |
| Army Research Office | W911NF-12-1-0594 | Rafael Yuste |
| Defense Advanced Research Projects Agency | N66001-15-C-4032 | Rafael Yuste |
| Defense Advanced Research Projects Agency | W91NF-14-1-0269 | Rafael Yuste |

The funders had no role in study design, data collection and interpretation, or the decision to submit the work for publication.

### Author contributions
Weijian Yang, Conceptualization, Resources, Data curation, Software, Formal analysis, Funding acquisition, Validation, Investigation, Visualization, Methodology, Writing—original draft, Writing—review and editing; Luis Carrillo-Reid, Conceptualization, Writing—review and editing; Yuki Bando, Data curation, Funding acquisition, Investigation, Writing—review and editing; Darcy S Peterka, Conceptualization, Resources, Software, Funding acquisition, Methodology, Writing—review and editing; Rafael Yuste, Direction, Conceptualization, Resources, Supervision, Funding acquisition, Project administration, Writing—review and editing

### Author ORCIDs
Weijian Yang (iD) http://orcid.org/0000-0003-0941-3496
Darcy S Peterka (iD) http://orcid.org/0000-0001-7351-5820
Rafael Yuste (iD) https://orcid.org/0000-0003-4206-497X

### Ethics
Animal experimentation: This study was performed in strict accordance with the recommendations in the Guide for the Care and Use of Laboratory Animals of the National Institutes of Health. All of the animals were handled according to approved institutional animal care and use committee (IACUC) protocols of Columbia University [protocol ID: AC-AAAM5100, AC-AAAM7951].

### Decision letter and Author response
Decision letter https://doi.org/10.7554/eLife.32671.018
Author response https://doi.org/10.7554/eLife.32671.019

## Additional files

### Supplementary files
• Transparent reporting form
DOI: https://doi.org/10.7554/eLife.32671.016

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
