## [Decision Letter]

[Editors’ note: a previous version of this study was rejected after peer review, but the authors submitted for reconsideration. The first decision letter after peer review is shown below.]

Thank you for submitting your work entitled "Three-dimensional Two-photon Optogenetics and Imaging of Cortical Circuits in vivo" for consideration by *eLife*. Your article has been reviewed by two peer reviewers, one of whom, Karel Svoboda is a member of our Board of Reviewing Editors and the evaluation has been overseen by a Senior Editor.

Our decision has been reached after consultation between the reviewers. Based on these discussions and the individual reviews below, we regret to inform you that your work will not be considered further for publication in *eLife*.

The reviewers appreciated the great potential of the photostimulation method. But the paper fell short in documenting the technical advances. The central claim of the paper is that the use of scanning galvos coupled with scanless SLM system for optogenetic stimulation greatly reduces the power required to photostimulate neurons relative to other 3D holography techniques. This was not directly shown. Moreover, optical and electrical crosstalk has not been characterized sufficiently.

Reviewer #1:

Previous work demonstrated the ability to manipulate specific cells with single-neuron precision within a given optical plane (2-dimensional stimulation). The aim of this work was to expand the number of addressable neurons by enabling stimulation in 3-dimensions using modest laser powers. The method (ability to stimulate 27 neurons in 3D) represents an advance in this field.

Major point:

The method portion that is novel is not thoroughly characterized.

Figure 1: Schematic of microscope and quantification of z-psf on photostimulation path. Important, z-psf is relatively broad due to the small size of the galvo, they could have made it tighter (at the expense of the size of the field-of-view). They also characterize targeting error and excitation efficiency relative to position in field of view. Overall, characterization of photostimulation path is thorough.

Spike count vs stimulation power and spiral duration. Spike latency, jitter and spatial resolution are all comparable to those reported from previous work that employed the same photostimulation technique (Packer, 2015). They need to provide more detail about how this measurement was done. How many different off-target spots were stimulated and where were they? What power and duration?

Why is there no quantification of excitation caused by the imaging laser? They do not mention the power used for imaging; in the discussion they state that they kept the imaging power "as low as possible".

Why are such long duration spirals used (up to 400 ms)? Excitation is maximized when the duration of the spiral is much less than the off time constant of the channel (40ms for C1V1).

Figure 2: They illustrate, as has been shown in previous work, that neurons that are targeted are reliably excited. The map highlights the effects of a broad z-psf. It would be nice to see if they could efficiently excite their targeted neurons with lower powers or durations while improving the spatial resolution.

Figure 3: The novelty is the ability to stimulate a 'large' number of neurons simultaneously. This capability is only demonstrated anecdotally in an experiment in which they simultaneously stimulate 27 neurons. The efficiency doesn't seem to drop off much going from 15 targeted neurons up to 27 targeted neurons. Would it have been possible to stimulate more? Can they move from 70% activation for the 27 neuron group up to 100% by changing stimulation parameters (i.e. power, duration, # of spirals)? Better characterization of the limits of the method is absolutely necessary.

Figure 4: To make this experiment more interesting it would be nice to demonstrate a result that could not be achieved by less sophisticated methods such as 1-photon stimulation. One such example would be to show how the change in selectivity in the population depends on the particular SOM neurons that were stimulated. What happens if another group of SOM neurons is stimulated?

Reviewer #2:

As stated by the authors in the Discussion section: "it has become the norm to use holographic approaches" for 2P stimulation of single or multiple opsin-expressing neurons. The central advance of this paper is the use of scanning galvos coupled with beamlets produced by a scanless SLM system for optogenetic stimulation. The authors claim this greatly reduces the power required to photostimulate neurons relative to other 3D holography techniques, which allows for further multiplexing of the beam and stimulation of more neurons. Unfortunately, this claim is not experimentally validated by the authors anywhere in the submitted manuscript. The manuscript suffers from a lack of rigor that is necessary to push the field forward and is poorly written.

1) Test the central claim of the paper: The microscope setup should be able to switch between scanless 3D holography, and the hybrid holographic-galvo spiral stimulation approach which the authors' claim greatly lowers the light powers required for stimulation. Both stimulation techniques could be used on the same animal with the same low-repetition laser. This would elegantly control for hardware and animal-to-animal variability (particularly opsin expression level). Please demonstrate that the light dosage required to stimulate neurons, either by GCaMP readout or cell-attached electrophysiology recordings (the latter would be better), is much smaller for the hybrid approach relative to scanless holography. I notice that the stimulation time is quite long for the spiral stimulation (the Materials and methods section has large range 10-2800 ms), so be sure to match time of stimulation as well as power to compare light dosage vs. excitability. In my opinion, this set of experiments is essential for the manuscript to be considered for publication.

2) It has now been several years that people have been publishing simultaneous 2P imaging and stimulation with GCaMP and C1V1. The authors report in the Results section "minimal cross-talk between the imaging and photostimulation beams" without characterization. If this is going to become a mainstream technique, it is vital that optical and electrical cross-talk be well characterized using these reagents. This is vital for progress in the field:

a) The authors' use cell attached recordings to measure latency and jitter of their stimulation paradigm, so they are proficient with in vivo electrophysiology. Please measure perturbations to cell activity (cell-attached) and membrane potential (whole cell) of C1V1 expressing cells when 2P scanning with typical conditions for GCaMP imaging. Please verify that the C1V1 cells that are recorded from also respond to the stimulation paradigm to confirm suitable expression levels.

b) The authors show an example for optical cross talk of the opsin-stimulation beam during GCaMP imaging (Figure 1—figure supplement 3), and their software solution. Please provide quantification for how large the artifacts tend to be, and how well the data processing removes it. One experiment to include would be to stimulate cells in a mouse with only GCaMP (no C1V1), and measure what percentage of cells, after implementing the data processing pipeline, are still characterized as stimulated by the activation criteria written in the methods. This will give a sense for false positives.

3) The authors report in the Results section "response rate of individual targeted cell remained high and stable (Figure 3, 82% +- 9%). Within a group of cells that were simultaneously photostimulated, the percentage of responsive cells was also stable (Figure 3, 82% +- 9%)."

a) I do not understand Figure 3. The stimulus conditions presented conditions go from 1-27 targets. The y-axis reports individual targeted cell response rate. What I imagine this graph is showing is the number of cells that were responsive during each of these conditions, yet that appears to be what Figure 3 is presenting. Is this graph showing excitation of each cell one at a time in all conditions (e.g. marching through each of 27 neurons in condition 7)? If so, what is the value of this graph? Figure 3 makes much more sense to me.

b) Figure 3 have very different distributions as a function of stimulation condition, so I find it hard to believe that the quoted percentage (82% +- 9%) is identical for both graphs.

c) I disagree with the assessment that the percentage of responsive cells is stable. The histogram peaks appear to generally decline with further multiplexing of the beam. I would eliminate this statement.

[Editors’ note: what now follows is the decision letter after the authors submitted for further consideration.]

Thank you for submitting your article "Three-dimensional Two-photon Optogenetics and Imaging of Cortical Circuits in vivo" for consideration by *eLife*. Your article has been reviewed by two peer reviewers, one of whom is a member of our Board of Reviewing Editors and the evaluation has been overseen by a Senior Editor. The following individual involved in review of your submission has agreed to reveal his identity: Bernardo L Sabatini (Reviewer #2).

The reviewers have discussed the reviews with one another and the Reviewing Editor has drafted this decision to help you prepare a revised submission.

Summary:

The manuscript has improved. The presentation is much clearer, and the benefits over and comparisons to previous techniques are better stated. The data is now better quantified. The techniques are more thoroughly explained and many previously missing details have been added.

Given that adding scan mirrors is a straightforward modification to most SLM based stimulation systems, this method could be readily adopted by many labs.

Essential revisions:

The comparison between scanning and scan-less approaches (Figure 1—figure supplement 2) is still scant. Given that many cells can be targeted at once, why is this analysis done for only 9 cells across 2 mice?

If the main goal of this paper is to maximize the number of neurons activated in some fixed time window, it is important to know how much faster cells can be activated using scanless techniques than scanning techniques. If scanless stimulation can activate cells significantly faster then, it might be a more efficient way to excite many cells in a fixed time window than the scanning techniques presented here.

It is nice to see a larger population activated than in the previous version. But what is the reason why so many cells fail to respond? Is it simply that they weren't expressing enough C1V1? This is worth checking via fluorescence of mCherry and discussing in the text.

Add some information about the depth of the cells in this analysis.

Regarding Figure 1—figure supplement 4. There appears to be some activation from scanning at 90mw relative to 0mw. A t-test should give a p-value around 0.05. The text should address this.

There are several relatively straightforward analysis steps that the authors should take to address the concerns regarding why cells that fail to respond during ensemble stimulation. Can the cells that fail to respond during ensemble stimulation be stimulated by themselves or are they just not excitable?

---

## [Author Response]

[Editors’ note: the author responses to the first round of peer review follow.]

Reviewer #1:1) Major point:The method portion that is novel is not thoroughly characterized.

Thank you for the critical comments and suggestions for the manuscript. In the revised manuscript, we present additional experiments to support the novel portion of our methods:

1) We demonstrated the simultaneous photostimulation of 83 cells across a relatively large cortical volume (480x480x150 μm^3^) in awake mice, using a total power of 300 mW with a photostimulation duration of 94 ms, while monitoring the neural activity in the network with GCaMP6f. We believe this represents an important advancement in the field. This result is reported in Figure 3 and Figure 3—figure supplement 1.

2) In our original manuscript, we discussed that our hybrid scanning approach which coupled scanning galvanometers with SLM hologram required a smaller power than other scanless approaches. We did not proof this experimentally. In the revision, we added a set of experiments for this comparison. We found that our approach took about half of the power to evoke a similar response in neurons when comparing with the scanless approach. This experiment supports our original claim. This result is reported in Figure 1—figure supplement 2.

3) We also performed a detailed characterization of the cross talk between imaging and photostimulation.

a) Imaging into photostimulation. We performed electrophysiological recordings and showed that our imaging conditions did not cause a significant increase of cell firing rates when comparing to “normal” spontaneous activity of these cells (both lasers off). This result is reported in Figure 1—figure supplement 4.

b) Photostimulation into imaging. We detailed various strategies in data processing to treat this cross-talk and think this is of practical utility to many readers. These discussions can be found in Discussion section and subsection “Data analysis”.

4) We documented the experiment parameters for each dataset and the statistical analysis used in each figure in detail.

2) Figure 1: Schematic of microscope and quantification of z-psf on photostimulation path. Important, z-psf is relatively broad due to the small size of the galvo, they could have made it tighter (at the expense of the size of the field-of-view). They also characterize targeting error and excitation efficiency relative to position in field of view. Overall, characterization of photostimulation path is thorough.Spike count vs stimulation power and spiral duration. Spike latency, jitter and spatial resolution are all comparable to those reported from previous work that employed the same photostimulation technique (Packer, 2015). They need to provide more detail about how this measurement was done. How many different off-target spots were stimulated and where were they? What power and duration?

1) In the previous version, this data was presented for single exemplar (which was stated in the figure caption). In the revision, we added in more data. The overall results stay the same (Figure 1, Figure 1—figure supplement 3). These results were obtained from cell-attached electrophysiology, which are stated in the corresponding figure captions.

2) For the spatial resolution of the photostimulation, we changed the photostimulation position versus the targeted cell position and read out the cell response either by cell-attached electrophysiology recording (Figure 1—figure supplement 3), or GCaMP6s (Figure 1—figure supplement 3). These two sets of experiments are done independently, and they show similar results. In Figure 1—figure supplement 3, we added an inset showing the experiment schematics. Furthermore, we labeled the off-target spots in C and E, so the readers can read out how many and where they are from the figure: for cell-attached electrophysiology recording, the number of off-target spots was 6 for lateral distance, and 10 for axial distance; for GCaMP6s recording, this number was 32 and 10 respectively.

3) We added the powers and duration for this experiment in the figure caption. In the revision, all these parameters have been stated in each experiment for each figure, as well as in the Materials and methods section.

In the revised manuscript, please see Figure 1, Figure 1—figure supplement 3. The experiment parameters are listed at the end of the caption in each figure.

In subsection “Photostimulation parameters”:

The pulse repetition rate of the photostimulation laser used in the experiment is 500 kHz or 1 MHz. […] This long photostimulation was composed of 175 continuous spiral scans, each lasting ~16 ms.

Why is there no quantification of excitation caused by the imaging laser? They do not mention the power used for imaging; in the discussion they state that they kept the imaging power "as low as possible".

The laser power used for resonant scanning imaging was typically below 50 mW and could be up to 80 mW for layers deeper than ~250 μm. We added this information in the Discussion section and Materials and methods section.

In addition, we performed electrophysiology recording and showed that our imaging conditions did not cause a significant increase of cell firing rates when comparing to “pure” spontaneous activity (all lasers off). This is shown in Figure 1—figure supplement 4.

In the revised manuscript, please see Figure 1—figure supplement 4 and in subsection “Microscope design”:

“The typical imaging power is <50 mW, and could be up to 80 mW for layers deeper than ~250 μm.”

Why are such long duration spirals used (up to 400 ms)? Excitation is maximized when the duration of the spiral is much less than the off time constant of the channel (40ms for C1V1).

We agree with the reviewer that spiral duration should be shorter to reduce the latency, jitter and increases the excitability, as shown in the characterization in Figure 1 and Figure 1—figure supplement 3. Depending on the aim of the experiments, we used different spiral durations:

1) Figure 2, and Figure 2—figure supplement 1. Long spiral duration was used to mimic a worst photostimulation condition. We investigated activation reliability and nonspecific activation effect.

2) Figure 3, where we show simultaneous photostimulation on large number of cells (>=50). We used the normal short spiral scan condition. The total photostimulation time was ~94 ms, which contained 5 continuous spiral scans, each lasting ~18.85 ms.

3) Figure 4, where we show SOM cell photostimulation during visual stimuli. The total duration of the visual stimuli was 2 seconds, and the total photostimulation time was 2.8 seconds, which contained 175 continuous spiral scans, each lasting ~16 ms.

We documented this at the corresponding figure captions, as well as in subsection “Photostimulation parameters”.

In subsection “Photostimulation parameters”:

“The pulse repetition rate of the photostimulation laser used in the experiment is 500 kHz or 1 MHz. […] This long photostimulation was composed of 175 continuous spiral scans, each lasting ~16 ms.”

3) Figure 2: They illustrate, as has been shown in previous work, that neurons that are targeted are reliably excited. The map highlights the effects of a broad z-psf. It would be nice to see if they could efficiently excite their targeted neurons with lower powers or durations while improving the spatial resolution.

Thank you for the suggestions. This figure showed single neuron activation across a 3D space. We moved this figure into supplement, as Figure 2—figure supplement 1. The psf of our photostimulation system is about 14.5 μm in FWHM (Figure 1). This translates into an ~40 μm FWHM for actual photostimulation (Figure 1—figure supplement 3), as confirmed by recording using both electrophysiology (Figure 1—figure supplement 3) and calcium imaging (Figure 1—figure supplement 3). This result was similar to that in Figure 2—figure supplement 1, though viewing in a slightly different perspective (nonspecific photostimulation). In these three sets of independent experiments, the photostimulation power and duration is listed below:

Figure 1—figure supplement 32.25 ~ 6 mW stim. power20 ms stim. durationFigure 1—figure supplement 33 ~ 4.5 mW stim. power154 ms stim. durationFigure 2—figure supplement 1~ 4 mW stim. power962 ms stim. duration

Though the parameter of the photostimulation varied quite a lot (specifically for Figure 2—figure supplement 1, where a worst photostimulation condition was mimicked), the situation for the axial resolution and nonspecific photostimulation looked similar. While we are working towards the replacement of the galvo, our thought is that the photostimulation beam could hit the dendritic arbors from other cells that course over the photostimulation region. This was clearly evident in the original slice work from our lab in Packer et al., 2012. This could play an important role in the nonspecific photostimulation. Somatic restricted opsins would likely help, and in fact have been shown to improve the resolution (Baker et al., 2016). We note that this has nothing to do with the underlying optical technology, but molecular biology, and we would expect our system to have similar improvements, while maintaining its unique and powerful large volumetric targeting capability. We are working on getting those constructs and viruses in our lab, and we acknowledged that the improvement of galvo and the application of future somatic restricted opsin would help in subsection “Nonspecific Photostimulation”.

In subsection “Nonspecific photostimulation”:

“In our current set of experiments, we use a relatively low excitation NA (~0.35) beam that is limited by the small mirror size (3 mm) of the post-SLM galvanometric scanners. […] A somatic-restricted channelrhodopsin 2 was reported recently (Baker et al., 2016), and showed reduced, but not eliminated, activation of non-targeted cells in vitro.”

4) Figure 3: The novelty is the ability to stimulate a 'large' number of neurons simultaneously. This capability is only demonstrated anecdotally in an experiment in which they simultaneously stimulate 27 neurons. The efficiency doesn't seem to drop off much going from 15 targeted neurons up to 27 targeted neurons. Would it have been possible to stimulate more? Can they move from 70% activation for the 27 neuron group up to 100% by changing stimulation parameters (i.e. power, duration, # of spirals)? Better characterization of the limits of the method is absolutely necessary.

Thank you for the critical comments. In the revision, this Figure 3 was moved to Figure 2. We performed additional experiments and presented them in the new Figure 3 and Figure 3—figure supplement 1. In Figure 3, the total number of simultaneous target cells are 40, 43 and 83. In Figure 3—figure supplement 1, the total number of simultaneous target cells is 50. Excluding the cells that never respond in all of the tested photostimulation patterns, the overall ensemble response rate (# of responsive cells / # of target cells) was 78% ± 7%, which is slightly worse than 82% ± 9% reported in Figure 2~27 simultaneous target cells).

In this new set of experiments, the photostimulation power was 3.6~4.8 mW/cell, and the photostimulation time was 94 ms, which contained 5 continuous complete spiral scans, each lasting 18.85 ms. In particular, the total power for the 83 cells case was 300 mW. The reduction of the photostimulation time reduced the ensemble response rate (data not shown). Although it would be possible to increase the power and photostimulation duration to increase the number of target cells as well as the ensemble response rate, we felt it better to limit the total power to values that have been shown not to cause heating damage with sustained illumination.

As for cellular “failures”, it is possible that when large number of pyramidal cells are targeted, they could also activate interneurons, which could in return suppress these pyramidal cells. The network dynamics pose a question on whether 100% activation rate can be achieved, and this could be a future study subject. It would require extensive electrophysiological studies and controls to test these hypotheses, and we believe these well beyond the scope of this paper.

In the revised manuscript, please see Figure 3, Figure 3—figure supplement 1.

Results section:

“A system that can modulate relatively large pools of neurons is desirable. With the low repetition-rate laser and hybrid scanning strategy (Discussion), the laser beam can be heavily spatially multiplexed to address a large amount of cells while maintaining a low average power. […] The nonspecific photoactivation becomes stronger for those cells being surrounded by the target cells, but overall it was confined within 20 μm from the nearest target cell (Figure. 3F).”

5) Figure 4: To make this experiment more interesting it would be nice to demonstrate a result that could not be achieved by less sophisticated methods such as 1-photon stimulation. One such example would be to show how the change in selectivity in the population depends on the particular SOM neurons that were stimulated. What happens if another group of SOM neurons is stimulated?

Thank you for the suggestions. In Figure 4, we simultaneously photostimulated 9 SOM cells, which has much better spatial specificity than typical one-photon stimulation. We do agree that it would be nice to show how the network reacts when different group of SOM cells are stimulated.

Indeed, we are working on this. Due to the setback of the availability of the SOM-cre animals, we would postpone this extended experiment as a future work dedicated to neuroscience study itself.

Reviewer #2:As stated by the authors in the Discussion section: "it has become the norm to use holographic approaches" for 2P stimulation of single or multiple opsin-expressing neurons. The central advance of this paper is the use of scanning galvos coupled with beamlets produced by a scanless SLM system for optogenetic stimulation. The authors claim this greatly reduces the power required to photostimulate neurons relative to other 3D holography techniques, which allows for further multiplexing of the beam and stimulation of more neurons. Unfortunately, this claim is not experimentally validated by the authors anywhere in the submitted manuscript. The manuscript suffers from a lack of rigor that is necessary to push the field forward and is poorly written.

Thank you for the critical comments and suggestions for the manuscript. In the revised manuscript, we presented additional experiments to support the argument that the hybrid scanning strategy requires less power than the scanless approach. We found that our approach took about half of the power to evoke a similar response in neurons when comparing with the scanless approach.

In addition, we demonstrated the simultaneous photostimulation of 83 cells across a relatively large cortical volume (480x480x150 μm^3^) in awake mice, using a total power of 300 mW with a photostimulation duration of 94 ms, while monitoring the neural activity in the network with GCaMP6f. We believe this represents an important advancement in the field.

Overall, our claim is that we extend the all-optical method (simultaneous imaging and photostimulation) from 2D to 3D, while being able to stimulate a relatively large number of cells in mice in vivo. To our knowledge, this is the first time to report two-photon photostimulation of such a large group of cells in 3D (83 cells across 150 μm depth range).

1) Test the central claim of the paper: The microscope setup should be able to switch between scanless 3D holography, and the hybrid holographic-galvo spiral stimulation approach which the authors' claim greatly lowers the light powers required for stimulation. Both stimulation techniques could be used on the same animal with the same low-repetition laser. This would elegantly control for hardware and animal-to-animal variability (particularly opsin expression level). Please demonstrate that the light dosage required to stimulate neurons, either by GCaMP readout or cell-attached electrophysiology recordings (the latter would be better), is much smaller for the hybrid approach relative to scanless holography. I notice that the stimulation time is quite long for the spiral stimulation (the Materials and methods section has large range 10-2800 ms), so be sure to match time of stimulation as well as power to compare light dosage vs. excitability. In my opinion, this set of experiments is essential for the manuscript to be considered for publication.

Thank you for the critical comments and suggestions. In the revised manuscript, we present a set of experiment to compare the power requirement for the spiral scanning approach and the scanless approach using pure hologram. We performed experiments on 9 cells over 2 mice in vivo. For each cell, we compared the evoked ΔF/F from 3 conditions: (1) Spiral scanning at 5 mW, 20 ms stimulation duration (2) Scanless disk at 5 mW, 20 ms stimulation duration (3) Scanless disk at 9 mW, 20 ms stimulation duration.

The spiral pattern and disk both covers the whole cell body, typically with a diameter of ~12 μm.

With one-way ANOVA test, we found that the evoked ΔF/F response has a significant difference between condition (1), (2), and condition (2), (3). Condition (1) has a trend of larger ΔF/F than condition (3), but not significant. We thus make an argument that it took about half of the power in the hybrid scanning approach than the scanless pure hologram approach to achieve a similar response in the same cells. This data set is presented in Figure 1—figure supplement 2.

In addition, we want to address the large range of stimulation time. There are three range of photostimulation time used in this manuscript: (1) Figure 2, and Figure 2—figure supplement 1. Long spiral duration (>400 ms but less than 1 second) was used to mimic a worst photostimulation condition. We investigated activation reliability and nonspecific activation effect. (2) Figure 3, where we show simultaneous photostimulation on large number of cells (>=50). We used the normal short spiral scan condition. The total photostimulation time was ~94 ms, which contained 5 continuous spiral scans, each lasting ~18.85 ms. (3) Figure 4, where we show SOM cell photostimulation during visual stimuli. The total duration of the visual stimuli was 2 seconds, and the total photostimulation time was 2.8 seconds, which contained 175 continuous spiral scans, each lasting ~16 ms.

We documented this at the corresponding figure captions, as well as in subsection “Photostimulation parameters”.

In the revised manuscript, please see Figure 1—figure supplement 2 and subsection “Rationale for our design: Minimization of laser power”:

“Regardless of the exact implementation, scanless approaches require higher laser powers per cell in general. About twice of the power is required in pure hologram compared with our hybrid strategy to achieve similar response in the same cells (Figure 1—figure supplement 2). It would likely require more power for the same excitation with temporal focusing as its tighter axial confinement would excite less of the membrane.”

In subsection “Photostimulation parameters”:

“The pulse repetition rate of the photostimulation laser used in the experiment is 500 kHz or 1 MHz. […] This long photostimulation was composed of 175 continuous spiral scans, each lasting ~16 ms.”

2) It has now been several years that people have been publishing simultaneous 2P imaging and stimulation with GCaMP and C1V1. The authors report in the Results section "minimal cross-talk between the imaging and photostimulation beams" without characterization. If this is going to become a mainstream technique, it is vital that optical and electrical cross-talk be well characterized using these reagents. This is vital for progress in the field:

*a) The authors' use cell attached recordings to measure latency and jitter of their stimulation paradigm, so they are proficient with* in vivo *electrophysiology. Please measure perturbations to cell activity (cell-attached) and membrane potential (whole cell) of C1V1 expressing cells when 2P scanning with typical conditions for GCaMP imaging. Please verify that the C1V1 cells that are recorded from also respond to the stimulation paradigm to confirm suitable expression levels.*

We performed cell-attached electrophysiology recording on C1V1 expressing cells when the imaging laser was scanning over the plane where the cell located, at different power conditions. We verified the recorded cells were responding to photostimulation both before and after this experiment. One-way ANOVA test show no significant difference on the cell firing rate between condition of 0 mW and 35~90 mW. Our typical imaging power was below 50 mW, though it could be up to 80 mW for layers deeper than ~250 μm. Furthermore, the scanning of the imaging laser cycles through different imaging planes (typically separated by ~50 μm each), leading to a 3~4 fold reduction of power depositing to the same plane. This measurement shows that the effect of the imaging laser into cell firing rate in our all-optical experiment is almost negligible. This is reported in Figure 1—figure supplement 4. Note that this result is more or less in line with those reported earlier, in Rickgauer et al., (2014) and Packer et al., (2015).

We did not perform the whole cell recording. It is certain that opsin activation produces subthreshold depolariazation in some cells, and we state as much. With respect to *firing*, and producing meaningful output that would drive network activity, we do not see considerable effect using our typical optical powers for imaging. As the focus of this paper is all optical interrogation, and the de facto standard of optical monitoring is calcium imaging, we use their established metrics of fluorescence transients to report the neural activity. These transients are generally indicators of action potentials, not sub-threshold signals, and we used the cell attached spiking rate to characterize the ground truth of these signals. We feel that this is well aligned, and consistent with the majority of imaging and photostimulation studies.

In the revised manuscript, please see Figure 1—figure supplement 4 and subsection “Minimizing cross-talk between imaging and photostimulation”:

“Cell-attached electrophysiology recording indicates that neuron firing rate has a trend to increase as the imaging laser power increases. However, we note that there is no significant difference of the firing rate under our typically volumetric imaging conditions (Figure 1—figure supplement 4), where the laser power was typically below 50 mW and could be up to 80 mW for layers deeper than ~250 μm. Nevertheless, as red indicators keep improving, we may see a future switch toward “blue” opsins again, as the spectral overlap between opsin and indicator can be reduced.”

b) The authors show an example for optical cross talk of the opsin-stimulation beam during GCaMP imaging (Figure 1—figure supplement 3), and their software solution. Please provide quantification for how large the artifacts tend to be, and how well the data processing removes it. One experiment to include would be to stimulate cells in a mouse with only GCaMP (no C1V1), and measure what percentage of cells, after implementing the data processing pipeline, are still characterized as stimulated by the activation criteria written in the methods. This will give a sense for false positives.

The cross-talk from opsin-stimulation into imaging was characterized as a sharp increase in fluoresce and then a sharp decrease (Figure 1—figure supplement 5), which is completely different from the dynamics of calcium transient. This cross-talk (fluorescence artifact) can be clearly identified by manual inspection.

In the revision, we provide various strategies to tackle the cross-talk, depending on the actual situation:

1) Baseline of GCaMP6 is weak and the fluorescence artifact is weak. This would not impact the analysis, so no extra data processing is required. This is the case for Figure 2, and Figure 2—figure supplement 1.

2) Fluorescence artifact is strong. In a typical operation condition, the photostimulation duration is short. If the fluorescent artifact only appears within 1 frame of data for all the imaging plane, we delete this frame, with negligible data loss. This is the case for Figure 3.

3) Fluorescence artifact is strong and photostimulation duration is long. We proceeded the data processing to remove the artifact, as described in the original manuscript. This is the case for Figure 4. We added additional figure panels to Figure 1—figure supplement 5, which describes the effect of this data processing. We also quantified the artifact suppression effect. We analyzed the raw ΔF/F of the cells (only expressed with GCaMP6s but not C1V1) right before photostimulation and right after the photostimulation onset. With the artifact suppression procedure, the overall raw ΔF/F across different cells stay similar before photostimulation and right after the photostimulation onset, though the standard deviation of the latter is larger. Nevertheless, the extracted events between these two periods stay similar. This is shown in Figure 1—figure supplement 5D-E. This shows the effectiveness of our method.

Finally, the constrained nonnegative matrix factorization algorithm used to extract the fluorescence signal could help, as it can identify the fluorescence artifact as part of the background and suppress it from the signal.

With all these procedures, the artifacts are not seen from the extracted fluorescence traces in Figure 2~4. All the above are documented in subsection “Minimizing cross-talk between imaging and photostimulation”.

The artifact size depends largely on the expression level of GCaMP6 (note: not C1V1). With a large number of target cells, the artifact could saturate the PMT. The above strategy provides a general guideline. As of which option to choose, one should always inspect the extracted fluorescence traces (by photostimulation trigger average) to determine whether there is artifact residue, and whether it would impact the subsequent data analysis. While the above procedures could suppress the artifacts in general, gated PMT which can shut down at the arrival of the photostimulation laser pulse could significantly help, as it could potentially facilitate a more unbiased analysis. The implementation of this hardware is currently in progress.

In the revised manuscript, please see Figure 1—figure supplement 5 and subsection “Minimizing cross-talk between imaging and photostimulation”:

“The second type of the cross-talk affects high fidelity recording of neural activity and is caused by fluorescence (or other interference) generated by the photostimulation laser directly, which would cause background artifact on the calcium signal recording during photostimulation. […] With all these procedures, the photostimulation artifacts are not seen from the extracted fluorescence trace in Figure 2~4.”

3) The authors report in the Results section "response rate of individual targeted cell remained high and stable (Figure 3, 82% +- 9%). Within a group of cells that were simultaneously photostimulated, the percentage of responsive cells was also stable (Figure 3, 82% +- 9%)."a) I do not understand Figure 3. The stimulus conditions presented conditions go from 1-27 targets. The y-axis reports individual targeted cell response rate. What I imagine this graph is showing is the number of cells that were responsive during each of these conditions, yet that appears to be what Figure 3 is presenting. Is this graph showing excitation of each cell one at a time in all conditions (e.g. marching through each of 27 neurons in condition 7)? If so, what is the value of this graph? Figure 3 makes much more sense to me.b) Figure 3 have very different distributions as a function of stimulation condition, so I find it hard to believe that the quoted percentage (82% +- 9%) is identical for both graphs.

We are sorry about this confusion. We explain the difference in the following discussion. Note in the revision, this figure was moved as Figure 2.

1) In Figure 2, we characterized the response rate of individual targeted cells. Multiple cells could be targeted together as an ensemble, but we were interested in the response rate in each individual cell (averaged across trials) within the targeted ensemble.

2) In Figure 2, we were interested in the trial-to-trial response rate in the ensemble level, i.e. the ratio between the number of responsive cells and the total targeted cell within an ensemble.

To be clear, we labeled “individual cell level response rate” for Figure 2 and “ensemble level response rate” for Figure 2. These two graphs characterize different aspects of the reliability of the photostimulation. The mean and standard deviation for condition 1 and 2 is the same in both graphs, as there is only one targeted cell in a targeted ensemble. Provided that the sample size is large enough, the overall mean and standard deviation should be very similar for each condition. This explains why the response rate for both individual cell level and ensemble level is 82% ± 9%. We specifically changed the wording in the manuscript and stated that both response rate is 82% ± 9% to avoid possible confusion. The different distribution in these two graphs comes from the fact that response rate for each cell in Figure 2 was averaged over trials, and we plotted the distribution across different cells. For Figure 2, each sample comes from a trial; that is why in condition 1 and 2, there were counts at 0% level. We indicated these difference in the figure caption.

In the revised manuscript, please see Figure 2 and the Results section:

“We further investigated the reliability of the photoactivation and also its influence on the activation of non-targeted cells – that is cells within the FOV not explicitly targeted with a beamlet. We performed 8~11 trials for each stimulation pattern. Cells not responding to photostimulation under any condition were excluded in this analysis (see Materials and methods section). We characterize the response rate at two levels: the individual cell (Figure 2) and the ensemble (Figure 2). The former characterizes the response rate of individual targeted cells in any stimulation pattern, and the latter characterizes the percentage of responsive cells within a targeted ensemble (defined here as ensemble response rate). As *M* increased, the response rate for both individual cell level and ensemble level remains high (both is 82% ± 9%, over all 7 stimulation conditions).”

c) I disagree with the assessment that the percentage of responsive cells is stable. The histogram peaks appear to generally decline with further multiplexing of the beam. I would eliminate this statement.

In the revised version, we eliminated this statement.

[Editors' note: the author responses to the re-review follow.]

The manuscript has improved. The presentation is much clearer, and the benefits over and comparisons to previous techniques are better stated. The data is now better quantified. The techniques are more thoroughly explained and many previously missing details have been added.Given that adding scan mirrors is a straightforward modification to most SLM based stimulation systems, this method could be readily adopted by many labs.

Thank you for the critical comments and valuable suggestions. By addressing the comments and implementing additional experiments and analysis, we believe the quality of the manuscript has improved.

Essential revisions:1) The comparison between scanning and scan-less approaches (Figure 1—figure supplement 2) is still scant. Given that many cells can be targeted at once, why is this analysis done for only 9 cells across 2 mice?

We apologize about the confusion and altered the caption and Title to be clear. In this figure, we photostimulate target cells one at a time. In this way, any non-specific photostimulation that could augment excitation between cells would be largely minimized. We believe this represents a clean comparison between the two approaches. To make it clear, the figure caption is modified from “Comparison between the spiral scan approach and the scanless approach using pure hologram” to “Comparison between the spiral scan approach and scanless (pure) holographic approach for single cell photostimulation.”

We also performed additional experiments and added more cells. The results are consistent with the original experiment and bolster the conclusion. Note that we re-labelled this figure to be Figure 2—figure supplement 1.

Furthermore, we performed additional experiments to compare these two approaches under different photostimulation durations, 20 ms, 10 ms, 5 ms, and 1 ms, reflecting a range of experimental conditions. The results are presented in the new Figure 2:

1) The new comparisons at 20 ms, 10 ms, and 5 ms were consistent with the original result with 20 ms photostimulation duration: the hybrid approach requires about half of the laser power than the scanless approach to evoke similar response in the neuron.

2) At 1 ms photostimulation duration, on the other hand, the hybrid approach shows a trend with smaller power budget (but not significant, p=0.17 using one-way ANOVA test) than the scanless approach.

We explained this result in the following.

1) The scanless approach employs a spatial multiplexed strategy, where the two-photon light is spatially distributed across the entire cell body; to maintain the two-photon excitation efficiency (squared-intensity) within its coverage area, a larger total power is typically required. The hybrid approach, on the other hand, is a combination of spatial (across different cells) and temporal (within individual cell) multiplexed strategy. While optimal strategy will depend on opsin photophysics, the opsin typically has a long opsin decay constant (10s of millisecond) and this favors the hybrid approach because the opsin channels can stay open during the entire (multiple) spiral scans. But at very short duration, the limited number of laser pulses per unit area may contribute to an efficiency drop of the hybrid approach versus scanless approach.

2) We added the above discussion in the Results section.

In the revised manuscript, please see Figure 2, and Figure 2—figure supplement 1.

In Results, paragraph 3:

“Compared with alternative scanless strategy like temporal focusing^11,16,21,22^ or pure holographic approaches^15^, where the laser power is distributed across the whole cell body of each targeted neuron, our hybrid approach is simple, accommodates large numbers of simultaneous targets, and appears to have a better power budget for large population photostimulation in general. […] But at very short duration, the limited number of laser pulses per unit area may contribute to an efficiency drop of the hybrid approach versus scanless approach.”

Figure 2, and Figure 2—figure supplement 1, caption: “Comparison between the spiral scan approach and scanless (pure) holographic approach for single cell photostimulation.”

2) If the main goal of this paper is to maximize the number of neurons activated in some fixed time window, it is important to know how much faster cells can be activated using scanless techniques than scanning techniques. If scanless stimulation can activate cells significantly faster then, it might be a more efficient way to excite many cells in a fixed time window than the scanning techniques presented here.

Thank you for the comments. We note that maximizing the total number of neurons activatable is a key thrust of this paper, but it is not the only one. Full, independent but simultaneous 3D targeting is another critical contribution. With regards to this concern, please see the response to comment 1. For photostimulation duration above 5 ms, the required power for spiral scan is about half compared to scanless approach to evoke similar response in neurons. At 1 ms photostimulation duration, the hybrid approach shows a trend with smaller power budget (but not significant, p=0.17 using one-way ANOVA test) than the scanless approach, for this particular opsin, but at peak intensity levels per cell we are not comfortable with. In this case, a higher laser repetition rate may facilitate the hybrid approach in this extreme time scale, as there would be more effective dwell time per unit area on the cell. Complete optimization of photostimulation at very short time scales is outside the scope of this paper, and we feel that the latencies and low jitter are already very good, and far better than one can “read-out” with calcium imaging.

3) It is nice to see a larger population activated than in the previous version. But what is the reason why so many cells fail to respond? Is it simply that they weren't expressing enough C1V1? This is worth checking via fluorescence of mCherry and discussing in the text.Add some information about the depth of the cells in this analysis.

In our experiments, we do not find a clear correlation between mCherry expression and the excitability of C1V1. This is consistent with the result reported in Packer et al., (2015). There are cells that expressing mCherry but do not appear to respond to single cell photostimulation. We do not see a specific correlation between the depth and the excitability.

In Figure 4, where photostimulation on large neuronal population is shown, we note that cells that could be photoactivated when the target ensemble is small may not get photoactivated when the size of the targeted ensemble increases. We have no direct measurements, but we suspect that when large number of pyramidal cells are targeted, they subsequently also activate interneurons, which could in return suppress these pyramidal cells. We have noted also a similar effect in brain slices. This network interaction is the subject of a future study.

We edited the revised paper to read (Results, paragraph 6):

“We then aimed to modulate relatively large groups of neurons in 3D. With the low-repetitionrate laser and hybrid scanning strategy (Discussion section), the laser beam can be heavily spatially multiplexed to address a large amount of cells while maintaining a low average power. […] We hypothesize that this could be due to feed forward inhibition, as targeted pyramidal neurons may activate local interneurons, which then could suppress the firing of neighboring cells. These network interactions will be the subject of future study. “

4) Regarding Figure 1—figure supplement 4. There appears to be some activation from scanning at 90mw relative to 0mw. A t-test should give a p-value around 0.05. The text should address this.

We added this t-test value in the figure caption.

We added the following sentence to the Figure 1—figure supplement 3 caption: “Paired-sample t-test between conditions of (0 mW, 35 mW), (0 mW, 55 mW), (0 mW, 90 mW) shows a p value of 0.50, 0.44, and 0.055 respectively.”

5) There are several relatively straightforward analysis steps that the authors should take to address the concerns regarding why cells that fail to respond during ensemble stimulation. Can the cells that fail to respond during ensemble stimulation be stimulated by themselves or are they just not excitable?

This is related to comment 3. In the large ensemble stimulation, we did not specifically perform single-cell photostimulation on each individual target cell. If cells fail to respond under any of the tested photostimulation patterns, we classified them as “cells never respond”, which is stated in the main text as well as in Materials and methods section. This could be the lack of C1V1 expression. As in the response in comment 3, we do see that cells that could be photoactivated when the target ensemble is small may not get photoactivated when the size of the target ensemble increases. It is possible that when large number of pyramidal cells are targeted, they could also activate interneurons, which could in return suppress these pyramidal cells. This network dynamics is deserved for a future study and is beyond the scope of this paper.